



# How does riming influence the observed spatial variability of ice water in mixed-phase clouds?

Nina Maherndl[1], Manuel Moser[2,3], Imke Schirmacher[4], Aaron Bansemer[6], Johannes Lucke[3,5], Christiane Voigt[2,3], and Maximilian Maahn[1]

[1]Leipzig Institute of Meteorology (LIM), Leipzig University, Leipzig, Germany
[2]Institute for Physics of the Atmosphere, Johannes Gutenberg University, Mainz, Germany
[3]Institute for Physics of the Atmosphere, German Aerospace Center (DLR), Wessling, Germany
[4]Institute for Geophysics and Meteorology, University of Cologne, Cologne, Germany
[5]Faculty of Aerospace Engineering, Delft University of Technology, Delft 2629, the Netherlands
[6]NSF National Center for Atmospheric Research, Boulder, Colorado, USA

**Correspondence:** Nina Maherndl (nina.maherndl@uni-leipzig.de)

**Abstract.**

Mixed-phase clouds (MPC) are a key component of the Earth's climate system. Observations show that ice water content (IWC) is not distributed homogeneously in MPC. Instead, high IWC tends to occur in clusters. However, it is not sufficiently understood, which ice crystal formation and growth processes play a dominant role in IWC clustering. One important ice growth process is riming, which occurs when liquid water droplets freeze onto ice crystals upon contact. Here, airborne measurements of MPC in mid- and high-latitudes are used to study spatial variability of ice clusters and investigate how this variability is linked to riming. We use data from the IMPACTS (mid-latitudes) and the HALO-(AC)[3] (high-latitudes) aircraft campaigns, where closely spatially and temporally collocated cloud radar and in situ measurements were collected. Ice cluster scales and IWC variability are quantified using pair correlation functions. By comparing IWC calculations accounting for riming to IWC calculations neglecting riming, we single out the influence of riming.

During all analyzed flight segments, riming is responsible for 66 % and 63 % of total IWC during IMPACTS and HALO-(AC)[3], respectively. In mid-latitude MPC, riming does not significantly change IWC cluster scales, but increases the probability of clusters occurrence. This enhancement occurs at similar scales as liquid water content variability. In cold air outbreak MPC observed during HALO-(AC)[3], riming impacts IWC clustering at two distinctive scales. First, riming enhances the probability of IWC clusters at spatial scales below 2 km, which corresponds to the wavelength of the roll cloud updraft and circulation features. Second, riming leads to additional IWC clustering at spatial scales of 3-5 km. We find that the presence of mesoscale updraft features leads to enhanced occurrences of riming and therefore additional IWC clustering. An increased liquid water path might increase the effect, but is not a necessary criterion. These results help to improve our understanding of how riming is linked to IWC variability and can be used to evaluate and constrain models of MPC.





# 1 Introduction

In mid- and high-latitudes, most precipitation stems from ice containing clouds (Mülmenstädt et al., 2015), which are a crucial component of the Earth's weather and climate systems. In mixed-phase clouds (MPC), ice particles and supercooled liquid
droplets coexist down to temperatures of about $-38\,°C$ in a thermodynamically unstable state. However, observations show that ice particles and liquid droplets are often mixed heterogeneously leading to the formation of hydrometeor clusters (Korolev et al., 2003; Field et al., 2004; Korolev and Milbrandt, 2022). Their spatial distribution as well as ice and liquid mass play a critical role not only in precipitation processes, but also cloud life time, radiative budget (Sun and Shine, 1994; Shupe and Intrieri, 2004; Turner, 2005), and climate feedbacks (Choi et al., 2014; Bjordal et al., 2020).

Quantifying spatial scales of cloud particle clusters has been the focus of previous studies. Most focused on liquid-phase clouds and analyzed liquid droplet clustering on small scales below $1\,m$ (Kostinski and Shaw, 2001; Shaw et al., 2002; Baker and Lawson, 2010), where turbulence plays a major role in clustering (Wood et al., 2005; Saw et al., 2012a, b). Studies looking at MPC suggest that ice clustering is present at different spatial scales (Korolev and Milbrandt, 2022; Deng et al., 2024). Deng et al. (2024) propose that ice clusters — defined as regions with enhanced ice particle number or ice water content — on larger
scales of a few $km$ dominate the inhomogeneity of the ice distribution. However, their analysis is based on one MPC case over China. It is unclear, if their findings are representative for different types of MPC. Additionally, which microphysical processes lead to ice water content (IWC) clustering at which scales is poorly understood.

In mid-latitudes, MPC can form during winter snowstorms. Snowfall produced by winter storms affects large populations, provides a vital freshwater source in many areas, but also contributes to accidents and economic losses. Snowfall is commonly
organized in banded structures and snowfall accumulations can show strong mesoscale variability. Even without orographic influences, snowfall amounts from a single storm can vary from a few millimeters to a meter over short distances (Picca et al., 2014; McMurdie et al., 2022). Larger-scale bands are likely caused by midlevel frontogenesis processes (Novak et al., 2004). Smaller-scale bands are likely linked to microphysical processes (i.e., primary and secondary ice formation, and ice growth processes) and therefore ice clustering in clouds. To what extent which processes play a dominate role is currently unclear. In
general, causes for the occurrence and evolution of snowbands at all scales remain poorly understood.

In the Arctic, MPC occur frequently (Shupe, 2011; Gierens et al., 2020; Moser et al., 2023; Kirschler et al., 2023) and have long lifetimes (Zuidema et al., 2005). During marine cold air outbreaks (MCAO), they form organized cloud streets consisting of shallow convective roll clouds, which can span distances over hundreds of kilometers (Müller et al., 1999). Organized patterns of increased ice water in-cloud and precipitation below-cloud can often be observed in Arctic MPC (e.g., Shupe et al.,
2008). In high latitudes, snowfall does not only impact freshwater supply, but also glacier mass balances and surface albedo.

Numerical forecast and climate models often fail to realistically predict or reproduce MPC properties, life-time and precipitation amounts (Morrison et al., 2020) for both Arctic MPC (Morrison et al., 2012; Ong et al., 2024) and winter storm clouds (Connelly and Colle, 2019; Harrington et al., 2013a, b; Jensen et al., 2017). One reason — that is common to all model calculations of MPC — is the uncertainty in representing ice particles. Ice formation and growth processes in MPC remain
poorly understood (Korolev et al., 2017) and their representations are therefore likely incomplete, even in sophisticated cloud



microphysics schemes (such as the new predicted particle properties (P3) scheme proposed by Morrison and Milbrandt, 2015). Gaps in our understanding of dominating ice processes hamper progression in representing MPC in models (Morrison et al., 2012). The misrepresentation of MPC and ice clouds has been suggested as large contributor to the uncertainty in CMIP6 climate model predictions (e.g., Bock et al., 2021).

To close these knowledge gaps, extensive observations of MPC properties are required. Aircraft field campaigns allow us to collect data in regions that are otherwise not (easily) accessible, like the Arctic. Airborne cloud radars provide information on the vertical cloud structure with larger spatial coverage than ground-based radars and higher spatial and temporal resolution as well as a smaller blind zone (Schirmacher et al., 2023) than satellite radars. Additionally, aircraft can be used to measure microphysical cloud properties in situ by flying through clouds. Although in situ cloud probes can provide reliable particle size

distribution (PSD) data (Korolev et al., 2013; Moser et al., 2023), accurate in situ measurements of IWC remain challenging (Heymsfield et al., 2010; Baumgardner et al., 2017; Tridon et al., 2019). In recent years, synergistic employment of both remote sensing and in situ instrumentation during airborne campaigns has become more common (Houze et al., 2017; McMurdie et al., 2022; Nguyen et al., 2022; Kirschler et al., 2023; Sorooshian et al., 2023; Wendisch et al., 2024; Maherndl et al., 2024). Combining collocated remote sensing and in situ observations has the great potential to improve remote sensing retrievals

(Blanke et al., 2023) and gain better insight on microphysical processes (Nguyen et al., 2022; Mróz et al., 2021). For example, by combining collocated radar and in situ particle size distribution (PSD) measurements, ice particle density can be inferred (Maherndl et al., 2024), which is often unconstrained in models.

Increased particle densities in MPC are commonly linked to riming, which describes the process of supercooled droplets freezing onto ice particles after contact. Riming is an important ice growth process by efficiently converting liquid to ice and

typically leads to increased particle mass, density, and fall speed (Heymsfield, 1982; Erfani and Mitchell, 2017; Seifert et al., 2019). Although riming can theoretically significantly increase IWC in MPC, it is unclear how much it contributes to IWC in reality and further to snowfall amounts on ground with different studies reaching different conclusions (Harimaya and Sato, 1989; Moisseev et al., 2017; Kneifel and Moisseev, 2020; Fitch and Garrett, 2022; Waitz et al., 2022).

In part due to our poor understanding of riming, it is often neglected in MPC studies, especially in low liquid water path

(LWP) conditions (Avramov et al., 2011; Yang et al., 2013; Oue et al., 2016). However, Fitch and Garrett (2022) observed shallow MPC with LWP below $50\,\mathrm{g\,m^{-2}}$ can produce heavily rimed particles. They proposed updrafts might be more important for the riming process than LWP. Neglecting riming in radar retrievals of IWC — i.e., by assuming mass-size and scattering relations developed for unrimed conditions — can lead to high retrieval biases (Tridon et al., 2019; Maherndl et al., 2023a).

In this study, we use airborne measurements of MPC in mid- and high-latitudes to study spatial scales of ice clusters and

investigate how IWC variability is linked to riming. We aim to:

1. Quantify spatial scales of ice clusters in MPC observed during the IMPACTS (mid-latitude winter storms) and HALO-(AC)[3] (Arctic MCAO clouds) aircraft campaigns.

2. Identify which spatial scales dominate the inhomogeneity of the ice distribution.

3. Characterize spatial scales at which riming enhances ice clustering and link to drivers of riming.





We use cloud radar and in situ cloud probe observations collected during the IMPACTS (McMurdie et al., 2022) and the HALO-(AC)$^3$ (Wendisch et al., 2024) aircraft campaigns. During both campaigns, two aircraft flew in an approximately vertically stacked coordinated pattern to collect spatially and temporally collocated radar and in situ data. The IMPACTS data was collected during four flights over the US East Coast and the Midwest. HALO-(AC)$^3$ data stems from three flights over the Fram Strait west of Svalbard. We use data collected during collocated flight segments to derive the normalized rime mass $M$ by closure of radar reflectivity and in situ PSD. Using two case studies, we present the "typical" cloud structures observed during IMPACTS and HALO-(AC)$^3$, before we present $M$ results for all collocated segments (Sect. 4.1). Then, we conduct a sensitivity test, to see how much the observed amounts of riming increase IWC (Sect. 4.2). Further, we quantify the spatial scales of ice particle number and IWC clusters using pair correlation functions (Sect. 4.3). By comparing IWC calculated with and without accounting for riming, we can infer at which spatial scales riming enhances the occurrence of IWC clusters. This approach allows us to tackle the above stated objectives leading to a better understanding of the spatial scales and drivers of ice clustering and thereby help to improve modeling of MPC.

## 2 Data

### 2.1 Airborne campaigns: IMPACTS and HALO-(AC)$^3$

The Investigation of Microphysics and Precipitation for Atlantic Coast-Threatening Snowstorms (IMPACTS, McMurdie et al., 2022) campaign was a NASA-sponsored field campaign to study wintertime snowstorms with a focus on precipitation variability in East Coast cyclones. Here, we use data collected during the winter 2020, where a variety of storms from the Midwest to the East Coast were sampled.

The DFG-funded field campaign HALO-(AC)$^3$ (Wendisch et al., 2024) took place in March and April 2022 and aims at investigating warm air intrusions and cold air outbreaks in the Arctic. In this study, we analyze data collected during MCAO conditions over the Fram Strait west of Svalbard.

Both aircraft campaigns have in common that collocated in situ and remote sensing measurements were conducted with two aircraft. During IMPACTS, the *ER-2* aircraft flew above clouds carrying a variety of passive and active remote sensing instruments including multiple frequency Doppler radars. Simultaneously, the NASA *P-3* aircraft collected measurements of microphysical cloud properties in situ while flying within clouds. During HALO-(AC)$^3$, the AWI aircraft *Polar 5* and *Polar 6* conducted measurements in a similar manner. *Polar 5*, equipped with a W-band radar among other remote sensing instruments flew above *Polar 6*, which carried out in situ measurements in clouds.

However, both campaigns cover different observation areas and spatial resolutions. With a typical flight speed of 200 (150) m/s the *ER-2* (*P-3*) covers a larger spatial scale at a coarser resolution than *Polar 5* and *Polar 6*, which fly at 60-80 m/s. While the *ER-2* and *Polar 5* flew at constant altitudes of 20 km and 3 km, respectively, *P-3* and *Polar 6* sampled at different altitudes up to 8.5 and 3 km, respectively. In this study, we investigate data collected during the flight days listed in Tab. 1. We selected these days because of the good collocation between the respective remote sensing and in situ aircraft as well as the data availability. Figure 1 shows all coordinated flight paths.





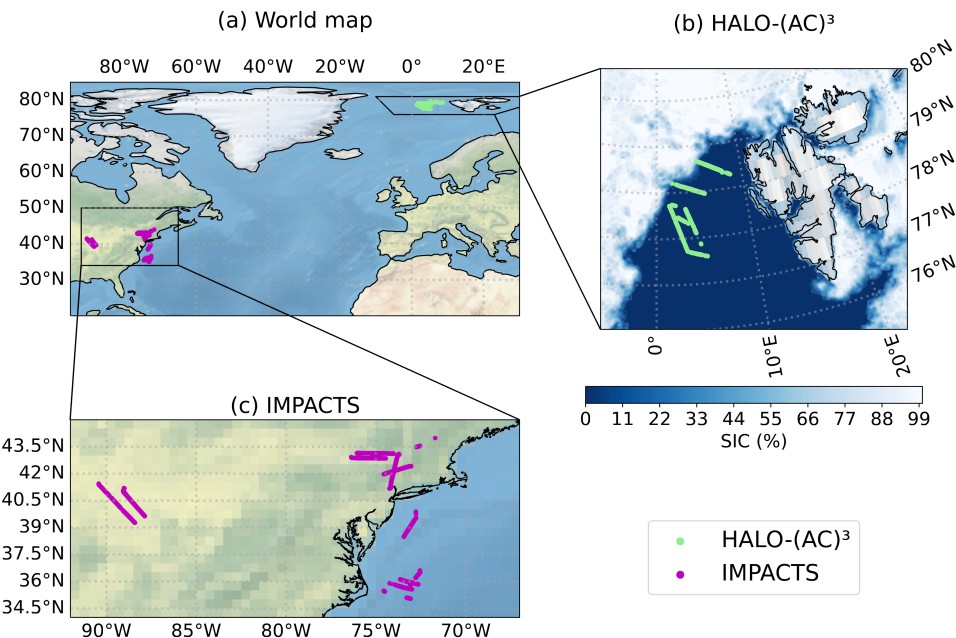

**Figure 1.** Flight tracks of (a) all analyzed coordinated flight segments, zoomed in on (b) HALO-(AC)$^3$, and (c) IMPACTS measurement area. In (b) the sea ice concentration (SIC) derived from the Advanced Microwave Scanning Radiometer 2 (AMSR2) onboard the GCOM-W1 satellite on 1 April 2022 is shaded in blue.

**Table 1.** Overview of analyzed flight days including campaign, measurement area, and synoptic situation.

| Campaign | Flight day | Measurement area | Synoptic situation / mission target |
|---|---|---|---|
| IMPACTS | 25 January 2020 | East Coast, New York | Warm occluded front |
| IMPACTS | 1 February 2020 | East Coast, Atlantic | Warm developing frontal system |
| IMPACTS | 5 February 2020 | Midwest | Shallow frontal zone |
| IMPACTS | 7 February 2020 | East Coast, Albany | Rapidly deepening cyclone |
| HALO-(AC)$^3$ | 28 March 2022 | Fram Strait | MCAO |
| HALO-(AC)$^3$ | 1 April 2022 | Fram Strait | MCAO |
| HALO-(AC)$^3$ | 4 April 2022 | Fram Strait | MCAO |

## 2.2 Instruments

Radar reflectivity $Z_e$ was measured by multiple radars during IMPACTS: X-band (9.6 GHz, EXRAD, Heymsfield et al., 1996, 2022), Ku and Ka-band (13.6 and 35.6 GHz, HIWRAP, Li et al., 2016, 2022), and W-band (94 GHz, CRS, McLinden et al., 2021, 2022). EXRAD consists of a nadir-pointing and a conically scanning beam, however, we only use the nadir-





pointing beam in this study. EXRAD, HIWRAP, and CRS sampled at 4 Hz, 2 Hz, and 4 Hz at vertical resolutions of 19 m, 26 m, and 26 m, respectively. During HALO-(AC)[3], a W-band radar (94 GHz, MiRAC-A, Mech et al., 2019; Mech et al., 2024a) was deployed. MiRAC-A was mounted with a 25° backwards inclination, sampled at 1 Hz and $Z_e$ data is available with

5 m vertical resolution. For both campaigns, $Z_e$ data is quality controlled and corrected for instrument orientation and aircraft motion (for MiRAC-A, see Mech et al., 2019). Uncertainties of $Z_e$ stemming from radar calibration are estimated to be below 1 dB and 0.5 dB for IMPACTS and HALO-(AC)[3] data, respectively. MiRAC-A $Z_e$ is corrected for attenuation due to liquid water content (LWC) as described in Maherndl et al. (2024); CRS $Z_e$ as described in Finlon et al. (2022). Attenuation due to water vapor and atmospheric gases is below 0.5 dB for all radars and therefore neglected.

During HALO-(AC)[3], brightness temperature $T_B$ measurements at 89 GHz were collected and are used to derive the LWP. Differences in $T_B$ for clear-sky and cloudy situations are used to retrieve LWP over ocean via a regression approach (Ruiz-Donoso et al., 2020; Maherndl et al., 2024).

Lidar measurement of backscattered intensities at 532 nm (parallel and perpendicular polarized) and 355 nm (not polarized; Stachlewska et al., 2010) are used to derive cloud top height (CTH) during HALO-(AC)[3] (Mech et al., 2022a; Schirmacher
et al., 2023; Maherndl et al., 2024; Mech et al., 2024b).

Cloud particle observations obtained with a variety of cloud probes cover a size range from 2 µm to about 2 cm for IMPACTS and 2.8 µm to 6.4 mm for HALO-(AC)[3]. For IMPACTS, we use data from a Cloud Droplet Probe (CDP, Lance et al., 2010) and a Fast-CDP (2-50 µm), a Two-Dimensional Stereo (2D-S, Lawson et al., 2006) probe (10-2000 µm, pixel resolution of 10 µm), one horizontally, and one vertically oriented High Volume Precipitation Spectrometer, version 3, (HVPS-3, Lawson et al.,
1998) probes (0.3-19.2 mm, pixel resolution of 150 µm). For HALO-(AC)[3], we use data from a Cloud Droplet Probe (CDP, 2.8-50 µm), a Cloud Imaging Probe (CIP, 15-960 µm, pixel resolution of 15 µm, Baumgardner et al., 2001), and a Precipitation Imaging Probe (PIP, 103-6400 µm, pixel resolution of 103 µm, Baumgardner et al., 2001). Here, we use merged particle size distribution (PSD) data from the respective campaign (Bansemer et al., 2022; Moser et al., 2023). As in Moser et al. (2023) and Maherndl et al. (2024), we assume all particles larger 50 µm in MPC to be ice crystals. LWC was measured in situ with a King
probe (King et al., 1978) and a Nevzorov probe (Korolev et al., 1998; Lucke et al., 2022; Lucke et al., 2024) during IMPACTS and HALO-(AC)[3], respectively. Due to poor data availability[1] and high uncertainties of IWC measurements, IWC is calculated from the PSD as described in more detail in Sect. 4. For more detail on IMPACTS and HALO-(AC)[3] instrumentation and data processing, we refer the reader to McMurdie et al. (2022) and Moser et al. (2023), Mech et al. (2022a), as well as Maherndl et al. (2024), respectively.

## 2.3  Synoptic situation

In this section, we give a brief overview of the "typical" synoptic situations encountered during the different field campaigns to provide context on the types of MPC that we analyze. We use one example flight segment for each campaign, which we describe in detail in Sect. 4.1.1 and 4.1.2.

---

[1]IMPACTS (2020): Water Isotope System for Precipitation and Entrainment Research (WISPER, Toohey et al., 2022) data product is available but unreliable under riming / icing conditions; HALO-(AC)[3]: Nevzorov probe data product only for April flights





During IMPACTS, observations of a variety of mid-latitude wintertime storms in different development stages were conducted. The focus was on observing banded precipitation structures. Observations range from a relatively weak and warm developing Atlantic low systems without major banding structures (1 February 2020) to rapidly deepening cyclones with significant snowfall and snowbands (5 February 2020). The majority of measurements stem from the U.S. Midwest, and close to the East Coast (both over ocean and land) ranging up to southern parts of Canada (Fig. 1). The coordinated *ER-2* and *P-3* flights on 5 February sampled an elevated warm front over shallow, pre-existing cold air as a low pressure center developed over Louisiana and Mississippi. The developing circulation around the low produced a low-level northeasterly flow across the Midwest. Due to the overrunning warm moist air from the south, precipitation in the form of rain (to the south) and snow (to the north) formed. During the period of observations, snowband structures were observed.

Measurements during HALO-(AC)[3] were conducted west of Svalbard over both open ocean and sea ice. However, clouds were very thin to non-existent over sea ice during all three flights used here. Northerly to northeasterly flow brought cold air masses from the sea ice of the higher Arctic to the comparatively warm open ocean. This led to the formation of roll cloud streets. During 1 April 2022 the MCAO was especially strong meaning the difference of the potential at sea surface and the potential temperature at 850 hPa was large (about 8 K), while during 28 March and 4 April 2022 weaker MCAO conditions were observed due to air masses being convected from North America over Siberia (28 March) or the central Arctic (4 April) to Svalbard (Walbröl et al., 2023).

## 2.4 Collocation

We use the same collocation criterion as in Maherndl et al. (2024), which is also extended to the IMPACTS data. To summarize, the nearest radar data point to the in situ measurements is selected following Chase et al. (2018) and Nguyen et al. (2022). Each 1 Hz, 2 Hz, or 4 Hz radar aircraft (*Polar 5* and *ER-2*) data point is matched with the spatially closest in situ aircraft (*Polar 6* and *P-3*) data point within a 5 min time window. We consider data with maximum spatial offsets of 5 km as "collocated". The closest radar range gate to the flight altitude of the in situ aircraft is chosen. Averaging over certain height ranges did not lead to significant improvements.

Rolling averages were applied to $Z_e$ and in situ data to obtain more robust statistics for the latter. To cover approximately the same spatial scales, averaging windows of 10 and 30 s are chosen for IMPACTS and HALO-(AC)[3], respectively. With typical flight speeds of 180-200 m/s and 60-80 m/s during IMPACTS and HALO-(AC)[3], respectively, this corresponds to spatial scales of 1.8-2.0 km and 1.8-2.4 km. We assume the in situ measurement is representative of the entire matched radar volume. Possible implications of this assumption on the riming retrieval are discussed in Maherndl et al. (2024).



## 3 Methods

### 3.1 Quantifying riming

We quantify riming using the two methods introduced in Maherndl et al. (2024). First, the combined method derives $M$
from a closure of in situ PSDs and collocated radar reflectivity $Z_e$. Second, the in situ method uses in situ measurements of
ice particle area $A$, perimeter $P$, and $D_{max}$ to derive $M$ for individual ice particles from which an average $M$ for the particle
population is derived. The methods in Maherndl et al. (2024) were developed for HALO-(AC)[3], but we apply the same methods
to IMPACTS data with slight adjustments due to different instrumentation. For IMPACTS, the combined method is applied
(separately) to X-, Ku-, Ka- and W-band $Z_e$ (see Sect. 4.1.3). As in Maherndl et al. (2024), we use the riming dependent
mass-size parameter relation for dendrites from Maherndl et al. (2023a). Dendrites were chosen, because 86.2 % of data during
the analyzed IMPACTS segments are within temperature ranges of -20 °C to -10 °C and -5 °C to 0 °C, where plate-like growth
of ice crystals is preferred (only 13.8 % of the data lie between -10 °C and -5 °C, where column-like growth dominates). We
assume dendrite shapes for the whole dataset, because of two reasons. First, Maherndl et al. (2024) found assuming plates
or dendrites gives the same results within uncertainty estimates, and second, we want to keep the analysis of IMPACTS and
HALO-(AC)[3] data as consistent as possible.

The in situ method is applied to 2D-S and HVPS-3 data for IMPACTS as was done using CIP and PIP data for HALO-
(AC)[3] in Maherndl et al. (2024). $P$ and $A$ measurements in pixel are used to calculate complexity $\chi = \frac{P}{2\sqrt{\pi A}}$. Synthetic rimed
aggregates are used to derive empirical functions relating $\chi$ and $D_{max}$ to $M$, where $\chi$ and $D_{max}$ are derived using the same
processing steps as for the respective cloud probes. Because these processing steps were slightly different for 2D-S and HVPS-3
operated during IMPACTS[2] than for CIP and PIP during HALO-(AC)[3], new fit functions (based on 18352 synthetic dendrites;
with $R^2 = 0.92$) had to be derived:

$$\log_{10}(M) = \frac{1.11 - \chi + 0.00141 \cdot D_{\max}}{0.00432 \cdot D_{\max} + 0.218}. \tag{1}$$

Only a subset of ice particles can be used to derive $M$ with the in situ method, because particles cannot touch edges to derive
$P$ and need to be large enough to derive meaningful $\chi$. We therefore assume the combined method — which uses the full PSD
— gives more reliable results. In situ method results are therefore only shown in Sect. 4.1.1 and 4.1.2 as references and the
combined method is used in all further analysis steps.

### 3.2 Characterizing scales of cloud variability

Similar to Deng et al. (2024), we use the pair correlation function (PCF) to quantify the spatial inhomogeneity of the observed
clouds. In discrete systems, the PCF describes the degree of deviation from the homogeneous Poisson process. In clouds, the
PCF can be used to quantify the degree of clustering or variability of a certain parameter such as the number concentration of
liquid droplets, the number concentration of ice particles, LWC, or IWC (e.g., Shaw et al., 2002; Saw et al., 2012a; Deng et al.,

---

[2]The number of perimeter pixel $P$ is computed by the sum of all pixel that are eroded when applying a "+" shaped erosion kernel without performing
dilation/erosion sequences as was done during HALO-(AC)[3].





2024). The PCF applied to a one-dimensional parameter $p$ is given by:

$$\eta(r) = \frac{\overline{p(0)p(r)}}{(\overline{p})^2} - 1, \tag{2}$$

where $p(0)$ is the parameter at a given point, $p(r)$ the parameter at the lag $r$ from that point, and $\overline{p}$ the average of $p$ (Kostinski
and Jameson, 2000; Shaw et al., 2002). Thus, $\eta(r)$ is a measure for the probability to find clusters of $p$ as a function of lag $r$
compared to $\overline{p}$. Positive values indicate the occurrence of clusters and the higher $\eta(r)$ the higher the probability to find clusters
at that scale. If $p$ follows a homogeneous Poisson distribution, which PCF assumes to be statistically homogeneous, $\eta(r) = 0$.
Negative values indicate that at the given scale, it is less likely to find clusters than on average over the whole segment.

In this study, only straight flight segments with a minimum of 200 s of continuous measurements are used to calculate $\eta(r)$.
We allow gaps with a maximum length of 5 s, which are linearly interpolated. Table 2 gives an overview of all segments we
analyze including duration and data amount.

Additionally, we use power spectra in order to gain insight on scales of variability of CTH and LWP during HALO-(AC)[3].
To do so, each data segment is mean-centered and linearly detrended. To minimize edge effects, a Hann window is applied to
each segment. Frequency is converted to wavelength using the aircraft speed $v_{air}$. With a minimum time range of 200 s per
segment, we capture spatial scales of 12 km for HALO-(AC)[3] meaning that we do not capture synoptic-scale motions. We
interpret results up to 0.1 Hz meaning spatial scales of 600 m.

Figure 2 visualizes the PCF and power spectra for synthetic data. In the case of a homogeneous Poisson process (Fig. 2a),
$\eta(r) = 0$ (Fig. 2d) and the power spectral density shows no significant peaks (Fig. 2g). For a periodic sine function with added
Poisson noise (Fig. 2b), $\eta(r)$ is positive for small lags and oscillates around 0 for lager lags with peaks occurring at multiples
of the wavelength $\lambda$ of the sine function (Fig. 2e). The power spectrum shows a peak at $\lambda$ (Fig. 2h). If the modulus function is
applied to the sine (Fig. 2c), $\eta(r)$ (Fig. 2f) is smaller than in Fig. 2e due to the lower signal to noise ratio and the oscillation
occurs at $\lambda/2$. The power spectrum also shows a peak at $\lambda/2$ (Fig. 2i).

### 3.3 Sensitivity study

To motivate our further analysis and to evaluate whether the retrieved amounts of riming significantly impact IWC, we conduct
a sensitivity study.

IWC is calculated by integrating the product of ice particle mass $m(D)$ and $N(D)$ from 0 to $\infty$

$$IWC = \int_0^\infty m(D)N(D)dD. \tag{3}$$

$m(D)$ is approximated by a power law relation with prefactor $a_m$ and exponent $b_m$

$$m(D) = a_m D^{b_m}, \tag{4}$$

where $D$ is the particle's maximum dimension and $a_m$ and $b_m$ are taken from Maherndl et al. (2023a) for varying amounts of
$M$. We assume that $N(D)$ follows a modified gamma distribution and use the normalized form introduced by Delanoë et al.



**Table 2.** Overview of analyzed segments including campaign, flight day, start and end times in UTC, and number of 1 s data points.

| Campaign | Flight day | Segment start | Segment end | Number of data points |
|---|---|---|---|---|
| IMPACTS | 25 January 2020 | 20:30:37 | 20:40:04 | 568 |
| IMPACTS | 25 January 2020 | 21:08:31 | 21:17:16 | 526 |
| IMPACTS | 25 January 2020 | 21:41:01 | 21:53:38 | 758 |
| IMPACTS | 1 February 2020 | 13:08:48 | 13:16:47 | 480 |
| IMPACTS | 1 February 2020 | 14:35:24 | 14:39:32 | 249 |
| IMPACTS | 5 February 2020 | 21:05:28 | 21:10:57 | 330 |
| IMPACTS | 5 February 2020 | 21:15:47 | 21:19:27 | 221 |
| IMPACTS | 5 February 2020 | 21:20:56 | 21:28:27 | 452 |
| IMPACTS | 5 February 2020 | 21:49:52 | 22:04:07 | 856 |
| IMPACTS | 5 February 2020 | 23:07:26 | 23:12:40 | 315 |
| IMPACTS | 7 February 2020 | 15:12:42 | 15:20:23 | 462 |
| IMPACTS | 7 February 2020 | 15:35:00 | 15:48:47 | 828 |
| IMPACTS | 7 February 2020 | 15:57:02 | 16:08:11 | 670 |
| HALO-(AC)³ | 28 March 2022 | 14:10:44 | 14:18:43 | 480 |
| HALO-(AC)³ | 28 March 2022 | 14:20:20 | 14:25:16 | 287 |
| HALO-(AC)³ | 28 March 2022 | 14:35:07 | 14:39:33 | 267 |
| HALO-(AC)³ | 28 March 2022 | 14:41:26 | 14:45:16 | 331 |
| HALO-(AC)³ | 1 April 2022 | 11:08:38 | 11:18:59 | 622 |
| HALO-(AC)³ | 1 April 2022 | 11:20:38 | 11:33:02 | 745 |
| HALO-(AC)³ | 1 April 2022 | 12:07:18 | 12:14:14 | 417 |
| HALO-(AC)³ | 1 April 2022 | 12:15:54 | 12:20:56 | 303 |
| HALO-(AC)³ | 1 April 2022 | 12:24:57 | 12:33:38 | 522 |
| HALO-(AC)³ | 1 April 2022 | 12:34:03 | 12:39:09 | 307 |
| HALO-(AC)³ | 4 April 2022 | 11:48:05 | 12:00:12 | 728 |
| HALO-(AC)³ | 4 April 2022 | 13:11:48 | 13:18:24 | 397 |
| HALO-(AC)³ | 4 April 2022 | 13:19:14 | 13:30:22 | 669 |

(2005, 2014) and extended by Maahn et al. (2015) for $D$ as the maximum dimension

$$N(D) = N_0^* \frac{(b_m + \mu + 1)^{b_m + \mu + 1} \Gamma(b_m + 1)}{\Gamma(b_m + \mu + 1)(b_m + 1)^{b_m + 1)}} \left( \frac{D}{D_m} \right)^{\mu} e^{-(b_m + \mu + 1)D/D_m}, \quad (5)$$

where $N_0^*$ is the overall scaling parameter, $\mu$ the shape parameter, and $D_m$ is the "mass-weighted" scaling parameter for the
250  particle size. We vary $N_0^*$ and $D_m$ — which can be calculated from PSD moments (see Maahn et al., 2015) — based on 10
to 90% quantile values derived from all measured PSDs during IMPACTS. Exclusively IMPACTS data was chosen, because
larger particles and higher number concentrations were measured during IMPACTS than during HALO-(AC)³. $\mu$ is varied from





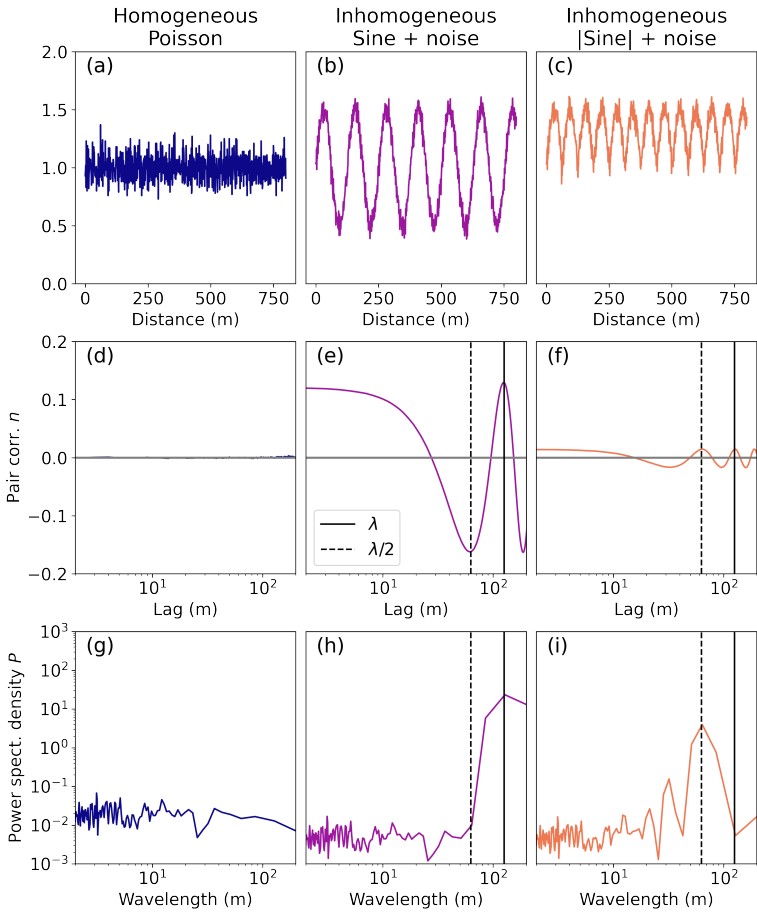

**Figure 2.** Schematic diagram introducing the pair correlation function (PCF) and power spectral density for (a) a homogeneous Poisson distributed signal, (b) a sine curve with wavelength $\lambda$ and added Poisson noise, and (c) the same sine curve but mirrored upwards along $x = 1$ to show the impact of $\lambda$ and signal-to-noise ratio. The respective PCF $\eta$ as a function of lag is shown in (d)-(f); the power spectra density as a function of wavelength in (g)-(i). The solid and dashed lines indicate $\lambda$ and $\lambda/2$ of the sine curve in (b).

0 to 64 based on extreme values reported in the literature (Tridon et al., 2022). $M$ is varied from 0.005 to 1, which correspond to the 10 % quantile of $M$ retrieval results from both campaigns and the maximum "physical" $M$ based on its definition. Results
are shown in Sect. 4.2.

## 4 Results and discussion

To characterize the influence of riming on the spatial variability of ice clusters, we first need to know the amount of riming as well as its impact on IWC and second spatial IWC cluster scales. Therefore, this section is structured as follows. First, we quantify the amount of riming observed during the two analyzed campaigns (Sect. 4.1). Then, we show that the retrieved





amounts of riming have a significant impact on IWC (Sect. 4.2). Finally, we quantify IWC variability (Sect. 4.3) and discuss the impact of riming on spatial scales and probability of IWC clusters.

## 4.1 Riming occurrence

Due to the different synoptic situations (Sect. 2.3) and measurement locations (Fig. 1), MPC properties vary between IMPACTS and HALO-(AC)[3]. Clouds during collocated IMPACTS segments have much larger vertical extents than during HALO-(AC)[3].

The median CTH during IMPACTS segments is 7.3 km (25-75 % quantile range: 6.3-7.8 km). Here, we define CTH as the height of the highest radar range gate with continuous $Z_e$ above the in situ aircraft altitude. Clouds observed during collocated HALO-(AC)[3] segments were predominately shallow roll clouds that formed during MCAOs. The maximum CTH during all segments was 2.2 km (25-75 % percentile range: 0.69-1.1 km). Cloud properties during 1 and 4 April are described in detail in Schirmacher, et al. (2024).

In the following, we give a brief overview on differences in MPCs encountered during the two campaigns using two "typical" example cases. We show a flight segment from 5 February 2020 for IMPACTS (Sect. 4.1.1), and from 1 April 2022 for HALO-(AC)[3] (Sect. 4.1.2). We present $M$, retrieved with combined and in situ method, and discuss uncertainties. Then, we extend to data from all collocated segments (Sect. 4.1.3).

### 4.1.1 Case study 1: Mid-latitude winter storm on 5 February 2020

Figure 3 shows a 64 km segment from 5 February, where *ER-2* and *P-3* were sampling a developing low pressure system over Illinois from 23:07:26 to 23:12:40 UTC. According to the level-2 MODIS cloud product (NASA worldview), the cloud top temperature (CTT) was $-33 \pm 5$ °C. W-band $Z_e$ shows the deep cloud with convective cell structures near cloud top from which sheared fall streaks stretch down (Fig. 3a). *P-3* measured number of ice particles larger 50 µm $N_i$ in the range 910 m³ to 2800 m³ (Fig. 3b). Here we show $D_{32}$ (Fig. 3b), which is the proxy for the mean mass-weighted diameter (e.g., Maahn et al.,

2015). $D_{32}$ is defined as the ratio of the third to the second measured PSD moments (e.g., Mitchell, 1996). During the first 20 km of the segment, ice particles had $D_{32}$ of about 3 mm and were lightly rimed with $M$ of about 0.02 (Fig. 3.c). Afterwards, $D_{32}$ increases up to 8 mm, indicating aggregates and $M$ drops below the riming threshold of 0.01. From $-88.9$°E onward, $D_{32}$ decreases and $M$ increases. Combined method $M$ results using the different frequencies show good agreement between X-, Ku-, and Ka-band. W-band results are likely biased high due to the high $D_{32}$ as will be discussed in Sect. 4.1.3. IWC is

calculated with Eq. 3 using (1) the measured PSD and mass-size parameters $a_m$ and $b_m$ for unrimed particles (blue line) and (2) $a_m$ and $b_m$ based on look up tables (Maherndl et al., 2023a) for each time step depending on retrieved $M$ for each frequency (black lines). The derived IWC from Ku-band $M$ varies between 0.015 gm⁻³ and 0.31 gm⁻³(panel 4). If riming is neglected, i.e., mass size parameter for unrimed particles are used in the IWC calculation, IWC is on average a factor of 3.7 lower.

The increase in $M$ starting at -88.7°E could be linked to the decrease in CTH (as seen by the radar). Some particles are

possibly rimed in liquid layers near cloud top and fall down to the measurement location. On their way down, they might undergo additional growth processes (condensational growth or aggregation) leading to a decrease of $M$, since $M$ is normalized to particle size. However, King probe measurements show that liquid water also occurs at the *P-3* position. Therefore additional



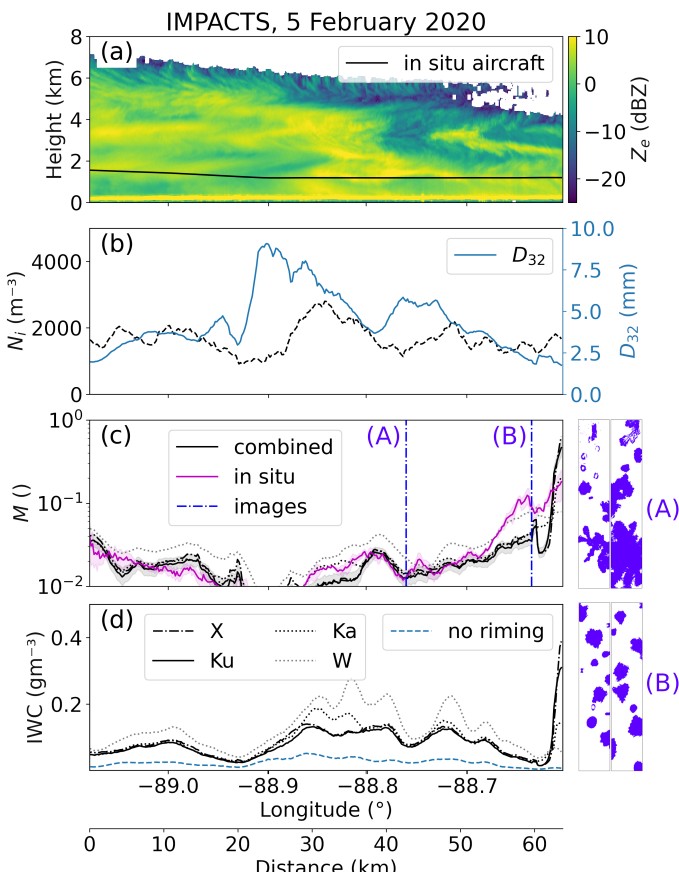

**Figure 3.** Collocated flight segment from 5 February 2020 at 23:07:26 to 23:12.40 UTC during IMPACTS. (a) W-band radar reflectivity $Z_e$, and *P-3* flight altitude; (b) ice number concentration $N_i$ and mass-weighted diameter $D_{32}$ derived from the 10 s running averaged particle size distribution (PSD); (c) normalized rime mass $M$ from combined (black) and in situ method (magenta) including uncertainty estimates (combined: optimal estimation (OE) standard deviation, in situ: 10 s running standard deviation), where the combined method was applied to X-, Ku-, Ka-, and W-band $Z_e$ (Ku-band results, which are used in the further analysis, are shown as solid lines); (d) ice water content (IWC) derived from the 10 s running averaged PSD and combined method $M$ (black) and assuming $M = 0$ (blue). Combined method results for different radar frequencies are drawn as dashed lines. 2-DS images at (A) -88.78°E and (B) -88.69°E are shown in blue next to panels (c) and (d).

riming can take place at the *P-3* location and possibly in cloud layers above. 2-DS images (Fig. 3) show a change from large, lightly rimed aggregates to small, more heavily rimed particles.

**4.1.2   Case study 2: Arctic roll clouds on 1 April 2022**

Figure 4 shows a 35 km segment from 1 April, where *Polar 5* and *Polar 6* were sampling perpendicular to the roll cloud structures formed during a MCAO over the Fram Strait from 11:20:38 UTC to 11:33:02 UTC (see  Maherndl et al., 2024,  for



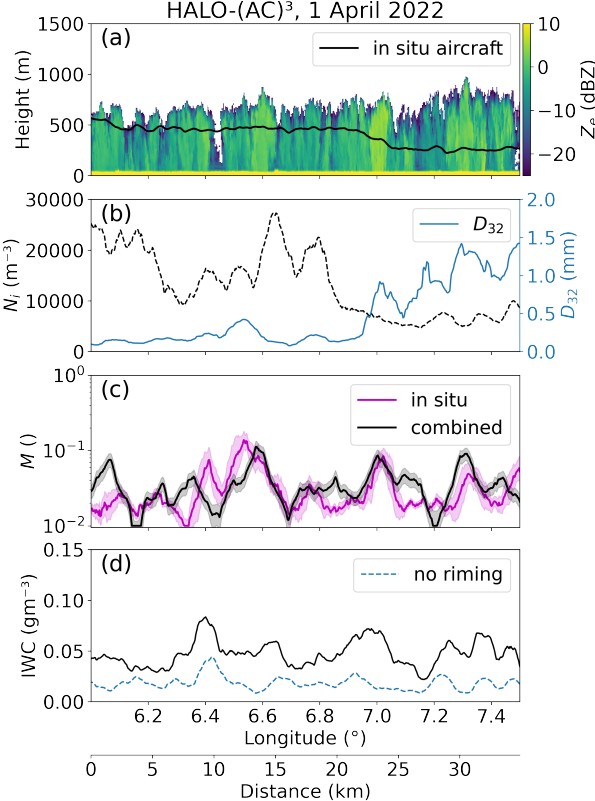

**Figure 4.** As in Fig. 3 but for the collocated flight segment from 1 April 2022 11:20:38-11:33:02 UTC during HALO-(AC)$^3$. Only W-band radar reflectivities are available.

a detailed discussion of the case as well as particle images). The MODIS CTT was $-18 \pm 5$ °C. W-band $Z_e$ shows the vertical structure of the individual cloud rolls (Fig. 4a). While *Polar 6* was flying close to cloud top, $N_i$ was high with a maximum

of 27300 m$^{-3}$, while $D_{32}$ was low with a minimum of 0.077 mm (Fig. 4b). Once *Polar 6* was descending, $N_i$ dropped to a minimum of 4600 m$^3$, while $D_{32}$ increased up to 1.4 mm (panel 2). $M$ oscillates between 0.01 and 0.1, with peaks occurring in streaks of high $Z_e$ (Fig. 4c). The resulting IWC is between 0.022 gm$^{-3}$ and 0.084 gm$^{-3}$. This is a factor 2.8 higher compared to using a mass-size parameterization for unrimed particles (Fig. 4d).

Both methods used to derive $M$ agree well for this segment in terms of $M$ distributions and location and extent of maxima

($R^2 = 0.52$). Statistical agreement between both methods was achieved during all HALO-(AC)$^3$ segments used in this study. However, spatio-temporal agreement could not be achieved for inhomogeneous cloud observations (e.g., when *Polar 6* was flying in and out of cloud close to CTH) as discussed in Maherndl et al. (2024).





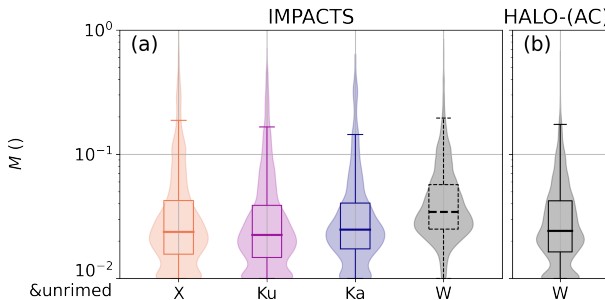

**Figure 5.** Box plots and superimposed violin plots showing normalized rime mass $M$ results obtained from a closure of collocated radar reflectivity $Z_e$ and in situ particle size distribution ("combined method" from Maherndl et al. (2024)) for radar reflectivities available during (a) IMPACTS and (b) HALO-(AC)[3]. W-band results during IMPACTS are dashed due to biases (see text). $M < 0.01$ are plotted at 0.01 to be visible on the logarithmic scale.

### 4.1.3 Campaign overview

In the previous section, two case studies were used to show differences between clouds observed during the two campaigns,
especially in terms of vertical extent, structure, and riming. In spite of these differences, normalized rime mass $M$ distributions derived for IMPACTS and HALO-(AC)[3] are similar (Fig. 5a, b). Median $M$ for all collocated IMPACTS segments are 0.024, 0.022, 0.025, and 0.034 when derived with X, Ku, Ka, and W-band $Z_e$, respectively. During collocated HALO-(AC)[3] segments, median $M$ is 0.024. For IMPACTS, the disagreement of the W-band results to the other frequency bands is due to the occurrences of large ice particle sizes. Due to saturation effects for $Z_e$ values associated to large particles at 94
GHz, the riming-dependent parameterization (Maherndl et al., 2023a) used here has a positive $Z_e$ bias for size parameters $x = 2\pi\alpha_e D_{max}/\lambda > 4$ where $x > 4$. Here $\alpha_e$ is the ice particle's effective aspect ratio, and $\lambda$ the radar wavelength. The positive $Z_e$ bias for $x > 4$ results in a positive bias of $M$. For IMPACTS, 25% of data have $D_{32} > 3.2$ mm which corresponds to $x = 4$ at 94 GHz assuming a typical value of $\alpha_e = 0.6$. Therefore W-band results for IMPACTS are not as trustworthy as the other wavelengths and are not used in the following analysis. Different to IMPACTS, the $M$ bias is negligible for HALO-
(AC)[3] due to the smaller particle sizes and $D_{32} < 3.2$ holds for 90% of the data. In Appendix A, an overview of microphysical parameter during each analyzed segment is given.

### 4.2 Sensitivity study

Although median $M$ are below 0.03 for both campaigns, even small amounts of riming — or rather changes in ice particle density — can result in large changes of IWC. Figure 6 shows IWC calculations assuming gamma PSDs with varying $N_0^*$ (left
column) and $M$ (right column) as a function of $D_m$. Similar to Maahn and Löhnert (2017), we find the shape parameter $\mu$ does not impact IWC or $Z_e$ significantly and therefore only $\mu = 0$ is shown. $D_m$, which can be seen as a proxy for particle size, has the largest impact on IWC. By changing $D_m$ from 1 to 8 mm, IWC changes by four orders of magnitude. IWC increases by about one order of magnitude, when $N_0^*$ — the proxy for total number concentration of particles — is increased by one order





of magnitude. Depending on $D_m$, varying $M$ can result in IWC changes up to two order of magnitudes. When only considering
$M$ values encountered during the analyzed campaigns, the change in IWC reaches one order of magnitude.

To show the impact of riming on radar reflectivity $Z_e$ — which can be seen as a proxy for IWC —, we conduct a sensitivity
study for Ku and Ka-band $Z_e$. In doing so, we aim to highlight the importance of accounting for riming in radar retrievals. $Z_e$
is forward simulated using the same PSDs with the Passive and Active Microwave radiative TRAnsfer tool (PAMTRA, Mech
et al., 2020) assuming a temperature of $-10\,°C$. Particle scattering is parameterized with the riming-dependent parameterization
(Maherndl et al., 2023a). X-band is not shown due to being nearly identical to Ku-band; W-band is not shown due to the riming-
dependent parameterization bias for large $D_m$ at W-band (see Sect. 4.1.3). Varying $M$ within observed ranges results in $Z_e$
changes of up to $20\,\mathrm{dB}$ depending on $D_m$ for both Ku- and Ka-band, albeit with slightly larger spread at Ka-band. Similar to
Fig. 6, varying $D_m$ results in the largest $Z_e$ changes. Observed ranges of $M$ result in larger $Z_e$ changes than observed ranges
of $N_0^*$. Therefore in our data set, $Z_e$ depends more heavily on riming than on number concentration.
We therefore conclude that even at low amounts of riming, as were observed during HALO-(AC)[3] and IMPACTS, the effect
of riming on IWC should not be neglected and can cause biases up to one order of magnitude.

## 4.3    Quantifying IWC variability

Because even small amounts of riming significantly impact IWC, we evaluate differences in IWC variability when accounting
for riming vs. when neglecting riming in the following. IWC is calculated with Eq. 3 based on the measured PSD and (1.) using
mass-size parameters $a_m$ and $b_m$ for unrimed particles ($\mathrm{IWC}_u$) and (2.) varying $a_m$ and $b_m$ for each time step as a function
of the retrieved $M$ ($\mathrm{IWC}_r$). During all analyzed IMPACTS flight segments, rime mass ($\mathrm{IWC}_r - \mathrm{IWC}_u$) makes up 68.6 / 65.7 /
68.8 % of $\mathrm{IWC}_r$ based on X- / Ku- / Ka-band results. During HALO-(AC)[3], rime mass makes up 62.7 %.

Figure 7 shows the average PCF $\eta$ over all analyzed IMPACTS and HALO-(AC)[3] segments for $N_i$ (Fig. 7 first column),
$\mathrm{IWC}_r$, and $\mathrm{IWC}_u$ (Fig. 7 second column). To visualize the difference between $\mathrm{IWC}_r$ and $\mathrm{IWC}_u$, Fig. 7, 3rd column shows the
$\eta_{IWC_r} - \eta_{IWC_u}$. By this, we can isolate the contribution of the riming process to IWC. Positive values of $\eta_{IWC_r} - \eta_{IWC_u}$
indicate riming increases the variability of IWC clusters at the given lag while negative values are related to riming smoothing
out IWC variability. Because we are interested at which spatial scales riming influences IWC variability, we only discuss the
differences larger zero.

Both in terms of $N_i$ and IWC, IMPACTS segments have higher $\eta$ on average than HALO-(AC)[3] segments meaning $N_i$ and
IWC have more variability on the investigated spatial scales (Fig. 7a, b). Note that both quantities are calculated from running
PSD averages of $10\,\mathrm{s}$ and $30\,\mathrm{s}$ for IMPACTS and HALO-(AC)[3], respectively, to cover similar spatial scales (about $1.8\,\mathrm{km}$)
given the different flight speeds. The smaller count of data points averaged for IMPACTS might lead to higher variability.
However, computing $\eta$ for $30\,\mathrm{s}$ running averages results in similar curves with close to the same lags where $\eta = 0$, and slightly
lower $\eta$, yet still higher than for HALO-(AC)[3] (not shown).
During IMPACTS, variability occurred at larger spatial scales than during HALO-(AC)[3] as indicated by positive $\eta$ at larger
lags (Fig. 7a, b). Differences between $\eta$ for $N_i$ and IWC indicate ice growth processes play a large role for IWC variability in
addition to ice formation processes. For both campaigns, $\eta > 0$ for IWC is shifted to larger spatial scales than for $N_i$ indicating





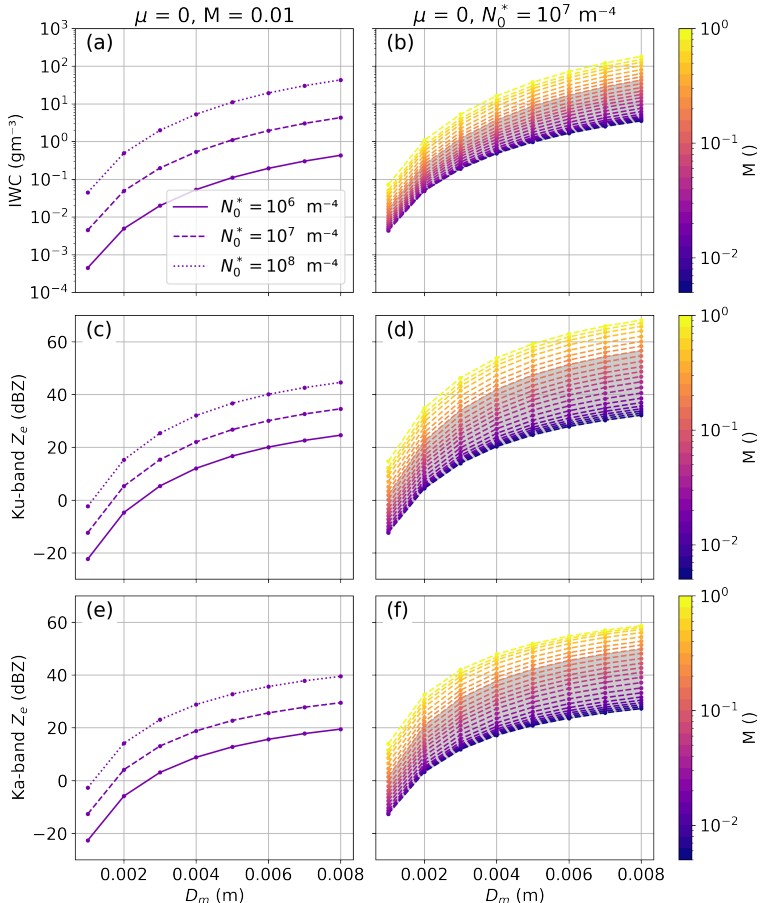

**Figure 6.** Ice water content (IWC) (top), Ku-band $Z_e$ (middle), and Ka-band $Z_e$ (bottom) calculated from gamma particle size distributions as functions of $D_m$ parameter. Results for varying $N_0^*$ parameter are shown as solid and dashed lines in (a), (c), (e); for varying normalized rime mass $M$ are color-coded in (b), (d), (f). Shaded areas in (b), (d), (f), (h) indicate $M$ ranges observed during IMPACTS.

ice growth processes lead to increased variability at large spatial scales. For IMPACTS, accounting for riming shifts scales of IWC variability to slightly smaller lags and increases $\eta$ significantly at small legs, meaning riming increases IWC variability at

lags < 5 km (Fig. 7c). For HALO-(AC)[3], riming leads to IWC variability at lags below 1 km as well as between 3-5 km. (Fig. 7c) However, differences between $\eta_{IWC_r}$ and $\eta_{IWC_u}$ are smaller than for IMPACTS.

### 4.3.1 Dependency on particle size

To identify which size range of particles contributes most to $N_i$ and IWC variability, we split the PSD into small ($50 < D_{max} < 300$ μm), medium ($300 < D_{max} < 900$ μm), and large ($D_{max} > 900$ μm) particle sizes to calculate $N_i$ and IWC (Fig. 7d-i).

For IMPACTS, the probability of small particle $N_i$ (IWC) clusters is higher than for medium and large particles below 3.5 km






**Figure 7.** Average pair correlation function (PCF) $\eta$ as a function of lag calculated for (a) ice number concentration $N_i$ and (b) ice water content (IWC) during IMPACTS (black) and HALO-(AC)$^3$ (green) segments. IWC is calculated with (solid line) and without (dashed line) accounting for riming and differences are plotted in (c). Shaded areas show standard deviations. In (d)-(i), the particle size distributions are split into small ($50 < D_{max} < 300$ μm), medium ($300 < D_{max} < 900$ μm), and large ($D_{max} > 900$ μm) particle sizes. (d)-(f) and (g)-(i) are as in (a)-(c) but showing size dependency of $\eta$ during IMPACTS and HALO-(AC)$^3$, respectively. Note the different y-axis scales.

(10 km). During HALO-(AC)$^3$, $\eta$ is similar regardless of size. However, positive $\eta_{IWC}$ — indicating the occurrence of IWC clusters — are shifted to slightly larger lags for large particles (9 km as opposed to 5-6 km for small and medium sizes).

The measurement location in-cloud could influence the dependency of $N_i$ and IWC variability on particle size due to size sorting, i.e., more small particles near CTH and larger particles at lower height. During the analyzed HALO-(AC)$^3$ segments, clouds were shallow and *Polar 6* measurements took place on average 440 m below the CTH (as measured by W-band radar).




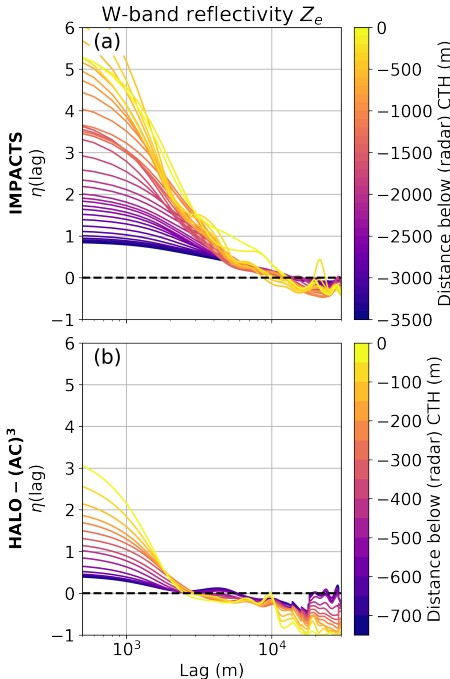

**Figure 8.** Average pair correlation function (PCF) $\eta$ as a function of lag calculated for horizontal cross section of W-band $Z_e$ (in linear units) during (a) IMPACTS and (b) HALO-(AC)$^3$ flight segments. Cross sections are taken in 100 m and 50 m steps from the average cloud top height (CTH) of each segment downward for IMPACTS and HALO-(AC)$^3$ data, respectively. Note the different colorbar scales.

During IMPACTS, much deeper cloud systems were observed and *P-3* sampled on average in larger vertical distances from cloud top (3.3 km) than during HALO-(AC)$^3$. W-band radar reflectivity $Z_e$ — which can be seen as a proxy for IWC — shows higher variability close to CTH for both IMPACTS and HALO-(AC)$^3$ clouds (Fig. 8). Similar to Fig. 7, we use PCF to characterize variability of $Z_e$. Here we use $Z_e$ in linear units such that there are no negative $Z_e$ values. For each IMPACTS

(HALO-(AC)$^3$) flight segment, $\eta$ is calculated for $Z_e$ cross sections in 100 m (50 m) steps from the average CTH downward. In general, $Z_e$ variability is larger close to cloud top at lags below 5 km and 2 km for IMPACTS and HALO-(AC)$^3$, respectively. The higher variability is likely linked to cloud top generating cells, which can be seen e.g., in case study 1 (Fig. 3a). Generating cells contain more liquid and ice and have stronger updrafts than adjacent cloud regions. HALO-(AC)$^3$ clouds show less variability and are homogeneous on smaller spatial scales ($\eta = 0$ is at smaller lags) than clouds during IMPACTS. Size sorting

might play a larger role for IMPACTS due to the larger cloud depths as opposed to the shallow MCAO clouds during HALO-(AC)$^3$. However, $N_i$ and $IWC$ distributions as functions of distance to CTH indicate the opposite (Appendix B). Nonetheless, $N_i$ and $IWC$ derived for small particles only show much more variability depending on the distance to CTH for IMPACTS (Appendix B).

  The higher variability of small particle counts during IMPACTS is therefore likely due to higher numbers of ice nucleating

particles (INP) available at mid-latitudes (Petters and Wright, 2015). During the analyzed HALO-(AC)$^3$ flight days, INP





concentrations collected with filters on board of *Polar 6* were very low, oftentimes below the detection threshold (Wendisch et al., 2024). No INP measurements were conducted during IMPACTS, therefore a direct comparison cannot be made. Another explanation could possibly be more secondary ice production (SIP) occurring during IMPACTS than during HALO-(AC)[3].

Differences between $\eta$ computed for $IWC_r$ and $IWC_u$ using the different size bins (Fig. 7f) show that riming enhances the probability of IWC clusters for lags smaller 9 km for small particles during IMPACTS. For medium and large particles, riming enhances IWC cluster probability at lags smaller 3 km. The enhancement is larger the smaller the lag for medium and large particles, whereas for small particles the largest enhancement is at a lag of about 2 km. An enhancement for small particles possibly hints at SIP connected to riming such as rime splintering. During HALO-(AC)[3] (Fig. 7i), riming enhances the probability of IWC clusters for lags smaller 4 km for small and medium particles and the enhancement is generally larger the smaller the lag. For large particles only lags of about 3-5 km lead to an enhancement of IWC variability.

### 4.3.2 Dependency on riming

To understand which spatial scales dominate the riming driven IWC variability, we conduct a Monte-Carlo random test for specific sampling distances following Deng et al. (2024). For each flight segment, we randomly select a sub-segment with a distance of $d$ km, where we vary $d$ in 1 km steps from 1 to 15 km. Then, we calculate $\eta$ for this segment. This is repeated 100 times and the average $\eta$ over all (sub)segments of the respective campaign is calculated. This approach allows us to first, handle the flight segments of different lengths in a statistically robust way, and second, analyze the dependence on flight segment distance. The results are shown in Fig. 9, where the average $\eta$ for $N_i$, $IWC_r$, and $IWC_u$ are plotted as functions of distance $d$ and lag. Curves (shaded) where $\eta = 0$ are included to show the maximum spatial scales at which ice clusters likely occur, given a sampling distance $d$.

During IMPACTS, the maximum $N_i$ cluster spatial scale increases from 0.6 km to 3.1 km at distances $d$ of 2 km to 15 km (Fig. 9a). King probe-measured LWC clusters behave similarly, increasing from 0.6 km to 3.0 km. This indicates simultaneous liquid and ice formation in regions with high supersaturation. Maximum IWC cluster scales (independent of accounting for riming or not) increase from 0.6 km to 3.6 km (Fig. 9b,c). At distances smaller 6 km, $N_i$ and IWC have about the same cluster scales; at distances larger 10 km, IWC clusters occur at larger spatial scales. Differences between $IWC_r$, and $IWC_u$ (Fig. 9d) reveal that riming enhances the probability of ice clusters for distances larger 6 km for lags from about 1 km to 10 km (at distances of 12 km). The enhancement occurs at similar spatial scales as LWC clusters, indicating riming is driven by LWC variability.

During HALO-(AC)[3], the maximum $N_i$ cluster spatial scale increases from 0.5 km to 3.7 km at distances of 2 km to 15 km (Fig. 9e). Similar to IMPACTS data, Nevzorov probe measured LWC clusters behave similarly, increasing from 0.5 km to 3.3 km, however having slightly smaller spatial scales. Maximum IWC cluster scales assuming no riming increase from 0.6 km to 3.8 km and therefore occur at about the same spatial scales as $N_i$ clusters (Fig. 9g). When accounting for riming, maximum IWC cluster scales show a distinct behavior for distances larger 10 km: $\eta$ increases at 3-5 km indicating that riming enhances variability on these scales (Fig. 9f). Differences between $IWC_r$, and $IWC_u$ (Fig. 9h), further highlight this feature.



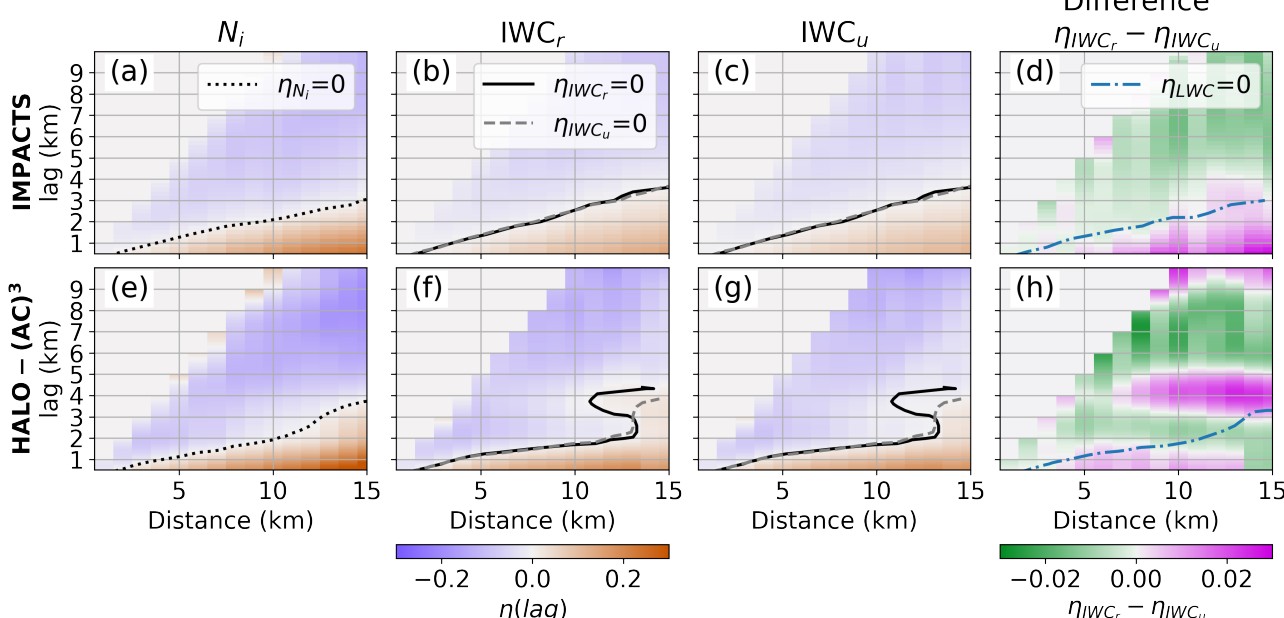

**Figure 9.** Average pair correlation function (PCF) $\eta$ as a function of distance and lag calculated using all (a-c) IMPACTS and (e-g) HALO-$(AC)^3$ flight segments for (a)&(e) $N_i$, (b)&(f) ice water content (IWC) accounting for riming $IWC_r$, and (c)&(g) IWC assuming no riming $IWC_u$. The Difference between (b) and (c) are shown in (d); difference between (f) and (g) in (h). $\eta = 0$ is drawn as shaded lines for the ice number concentration $N_i$ (dotted black), $IWC_r$ (solid black), $IWC_u$ (dashed grey), and liquid water content (LWC, dash-dotted blue), where LWC measurements from King (Nevzorov) probe measurements obtained during IMPACTS (HALO-$(AC)^3$) are used.

To explain the different spatial scales where riming enhances IWC variability, we look at lidar derived CTH. In previous
sections, we derived CTH from radar measurements to make IMPACTS and HALO-$(AC)^3$ comparable. During HALO-$(AC)^3$
a more sophisticated CTH product based on lidar — which is more sensitive to liquid layers at cloud top than the radar — is
available and used the following. The lidar detects small liquid droplets at cloud top, which follow vertical motions, therefore
leading to higher CTH in updraft regions. When computing the average power spectrum of CTH observed during the studied
flight days, distinct peaks at wavelengths of 750 m and 1.2 km occur for all days, which corresponds to the typical roll cloud
and circulation wavelengths as derived by Schirmacher, et al. (2024) (Fig. 10a, d, g). At these wavelengths, peaks in LWP
also occur for all days (Fig. 10b, e, h) further indicating enhanced formation and growth of liquid droplets in the updraft
regions of the convectional cell cloud structures. On 28 March, a distinctive peak in the CTH spectrum at 3-5 km indicates
additional mesoscale updraft features (Fig. 10a). However, the LWP spectrum only shows a weak peak towards 5 km (Fig.
10b). On 1 April, both CTH and LWP power spectra have peaks at 3-5 km (Fig. 10d,e). On 4 April, no peak distinctive peaks
at wavelengths of 3-5 km are visible (Fig. 10g,h). Given that the least (most) amount of riming occurred on 4 (1) April, we
conclude that in the studied MCAO clouds mesoscale updraft features likely enhance riming at spatial scales of 3-5 km. The




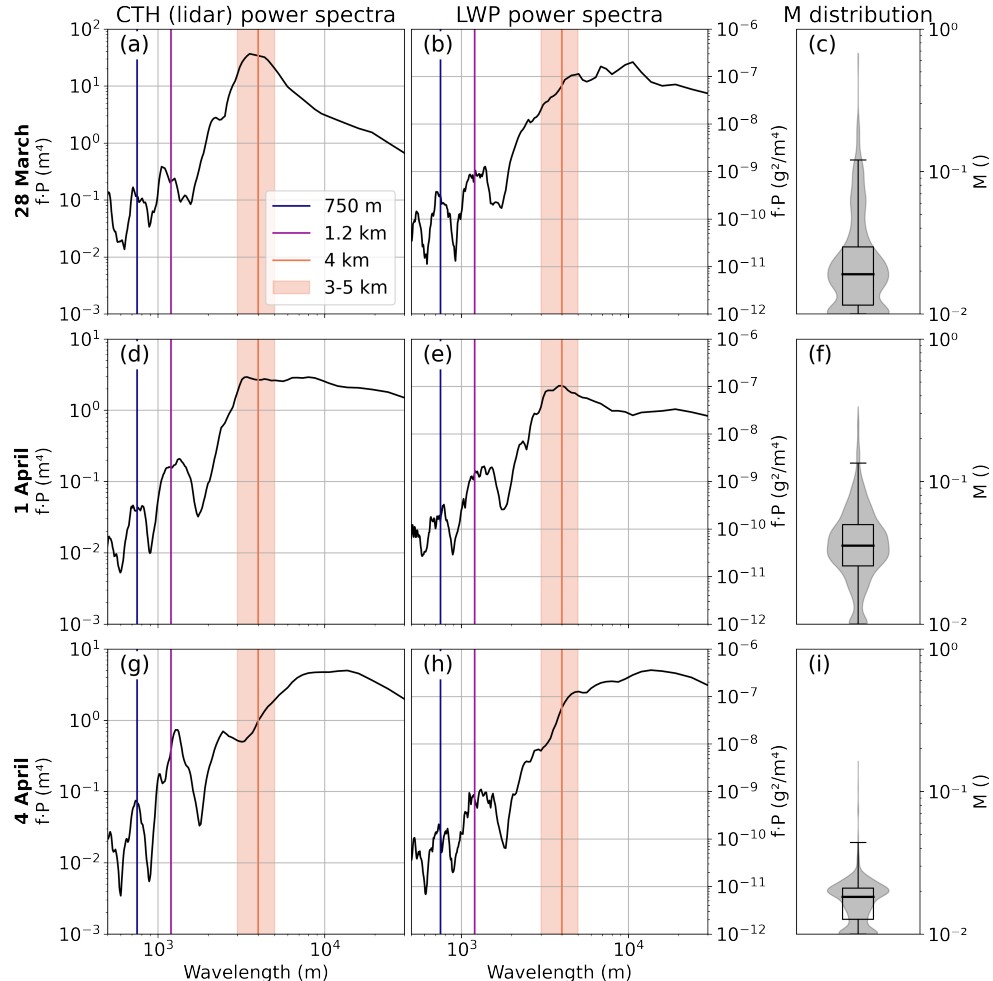

**Figure 10.** Power spectra of (a), (d), and (g) cloud top height (CTH) as derived from lidar and (b), (e), and (h) liquid water path (LWP) during collocated HALO-(AC)³ flight days. The wavelength has been calculated based on the aircraft flight speed. The blue and purple lines show the typical roll cloud and circulation wavelengths as derived by Schirmacher, et al. (2024). The orange shaded area shows the 3-5 km range, where riming causes additional IWC clustering. (c), (f), and (i) show the corresponding normalized rime mass $M$ distributions.

enhancement could occur either due to prolonged lifetimes of ice crystals in clouds (28 March) or increased amounts of liquid water or both (01 April) and leads to an increase in IWC amount and variability.

## 4.4 A conceptual model of how riming impacts IWC clusters in MCAO roll clouds

The results discussed above help to better understand scales of IWC clustering in different types of MPC and link to some microphysical processes involved. Although there are substantial unknowns, the following summarizes our findings from the perspective of collocated remote sensing and in situ measurements.





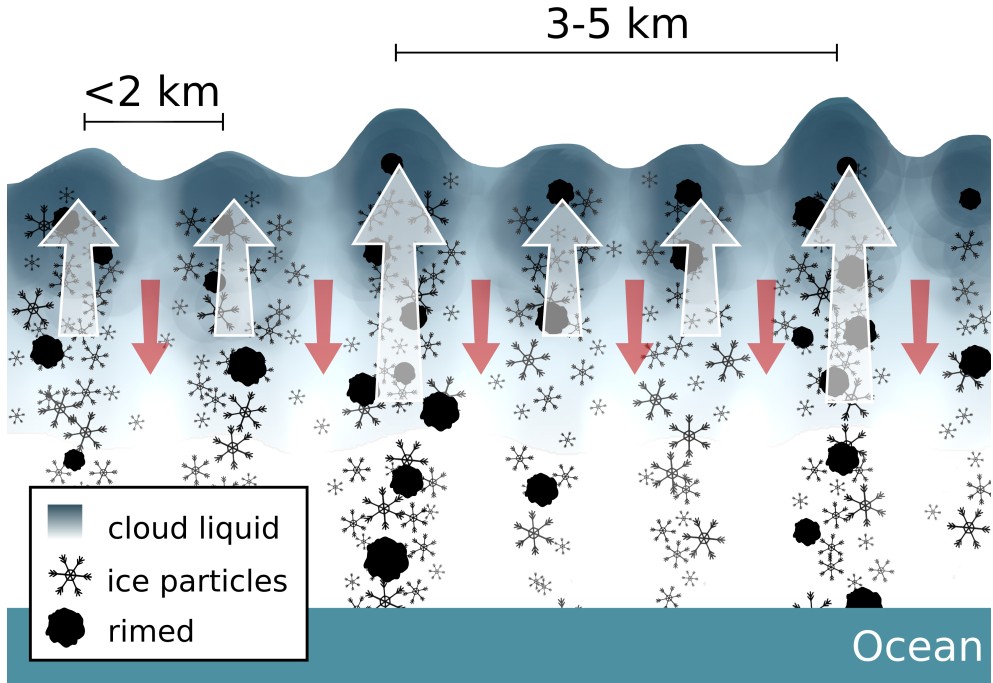

**Figure 11.** A conceptual diagram summarizing ice cluster spatial scales driven by riming as observed in MCAO roll clouds during HALO-(AC)[3]. For further explanations see text.

Figure 11 shows a sketch of the maximum spatial scales, where we found ice clusters to occur for MPCs observed during MCAOs during HALO-(AC)[3]. In these MCAO roll clouds, ice clusters occur at spatial scales of the roll cloud wavelengths. In the updraft regions of the convectional cells, which occurred on average every 750 m and 1.2 km, liquid droplets and ice particles are formed. LWP and CTH are increased due to the vertical motions and liquid condensation. Ice particles grow through condensational growth and riming, which leads to enhanced probabilities of ice clusters at these scales. When ice particles' masses have increased sufficiently, they precipitate or might sublimate below cloud. Aggregation might occur as ice particles collide. In the presence of additional mesoscale updraft features, IWC clusters also occur at spatial scales of 3-5 km. Due to the stronger vertical motion, ice particles are suspended longer, have more time to rime and can reach higher masses before precipitating. Increased LWP might enhance the amount of riming, but is not a necessary criterion based on the analyzed cases. This hypothesis is supported by the fact that the observed LWP is not sufficient to explain the retrieved rime masses assuming particles continuously collecting liquid water by falling through the liquid layer as we show in Appendix C. The enhanced occurrence of riming drives the additional increase of IWC cluster probability at spatial scales of 3-5 km.



## 5 Conclusions

In this study, airborne measurements of mixed-phase clouds (MPC) in mid- and high-latitudes are used to study spatial variability of ice clusters. We further investigate how this variability is linked to riming, which we quantify by closure of collocated cloud radar reflectivity and in situ particle size distribution (PSD) measurements. Pair correlation function (PCF) is used to quantify ice cluster scales and ice water content (IWC) variability when first, accounting for riming ($IWC_r$), and second, neglecting riming ($IWC_u$). The main findings are as follows:

1. Although synoptic situations and the resulting clouds systems were vastly different during the two analyzed aircraft campaigns, the retrieved amounts of riming were similar with median normalized rime masses $M$ of 0.023 and 0.024 during IMPACTS (mid-latitude winter storms) and HALO-(AC)³ (Arctic MCAO roll clouds) segments, respectively (Fig. 5). Clouds were deep (shallow) during IMPACTS (HALO-(AC)³) segments and in situ measurements were conducted on average in vertical distances of 3.3 km (440 m) to cloud top.

2. The observed spread of $M$ can increase IWC up to two orders of magnitude, depending on the size of the particle population (Fig. 6). In sum, rime mass makes up about 66 % and 63 % of total IWC during the analyzed IMPACTS and HALO-(AC)³ flight segments, respectively. Therefore, riming has a similar impact on IWC as the observed spread of number concentration and should not be neglected when estimating IWC.

3. PCF revealed that $N_i$ cluster occur with increased probability on spatial scales smaller 10.5 km and 6.5 km during IMPACTS and HALO-(AC)³, respectively. IWC clusters dominate for spatial scales of 10 km and 7 km. During IMPACTS, small particles dominate $N_i$ and IWC variability on small spatial scales, whereas there is no dependence on particle size during HALO-(AC)³ (Fig. 7). This could be linked to ice formation processes and the higher availability of INP at mid-latitudes. However, this hypothesis could not be confirmed with the available data.

4. During IMPACTS, maximum $N_i$, IWC and LWC cluster spatial scales are 0.6 km for distances of 2 km and increase to about 3 km for distances of 15 km. During HALO-(AC)³, maximum $N_i$, IWC and LWC cluster spatial scales are similar with about 0.5 km for distances of 2 km and about 4 km for 15 km. However, during HALO-(AC)³ IWC cluster probability is increased on scales of 3-5 km when segment distances are larger 10 km (Fig. 9).

5. During IMPACTS, accounting for riming does not significantly change IWC cluster scales, but increases the probability of clusters for segment distances larger 6 km (Fig. 9d). This enhancement occurs at similar scales as LWC variability. More riming likely occurs in regions with enhanced concentration of liquid water and increases IWC. Since clusters of IWC neglecting riming have similar spatial scales as $N_i$, LWC, and IWC accounting for riming, ice clustering is likely linked to ice formation processes in regions of high supersaturation with respect to liquid and ice .

6. In contrast, riming impacts IWC clustering at two distinctive scales during HALO-(AC)³ (Fig. 9h). First, riming enhances the probability of IWC clusters at spatial scales below 2 km, which corresponds to the wavelength of the roll cloud updraft features. $N_i$, $IWC_r$, $IWC_u$, and LWC all have similar spatial variability indicating simultaneous ice and liquid



formation and growth in this regions. The enhanced concentrations of liquid again enhance riming, which increases IWC. Second, riming leads to IWC clustering at spatial scales of 3-5 km, which cannot be explained by the typical roll cloud and roll circulation wavelengths. Power spectra of CTH show peaks at these spatial scales on the flight days with enhanced riming (Fig. 10). This indicates that the presence of mesoscale updraft features — which cause higher lifting of small particles near cloud top and therefore increased CTH — leads to enhanced occurrence of riming and therefore additional IWC clustering. Increased LWP might increase the effect, but is not a necessary criterion based on the analyzed days. Theoretical analysis shows that updrafts are likely required to explain the observed riming values (Fig. C1).

These results help to improve our understanding of how riming is linked to IWC variability and can be used to evaluate and constrain MPC models. While we have shown that riming enhances in-cloud IWC variability and causes additional IWC clustering at large spatial scales of 3-5 km in Arctic MCAO clouds, further research is needed to link these findings to surface precipitation. Future studies should investigate the link between riming-driven IWC variability and snowfall variability. In addition, profiles of vertical wind speed and turbulence are needed to better understand their importance for riming.

*Data availability.* Processed in situ (https://doi.org/10.1594/PANGAEA.963247, Moser et al., 2023), Nevzorov probe (https://doi.org/10.1594/PANGAEA.963628, Lucke et al., 2024) and MiRAC-A data (https://doi.org/10.1594/PANGAEA.964977, Mech et al., 2024a) as well as AMALi CTH (https://doi.org/10.1594/PANGAEA.96498, Mech et al., 2024b) from the HALO-(AC)[3] campaign are available on PAN-GAEA. The IMPACTS data (https://doi.org/10.5067/IMPACTS/DATA101, McMurdie et al., 2019) and the individual datasets cited within this paper can be found at the NASA Global Hydrology Resource Center's DAAC. The data set of simulated rimed aggregates generated for Maherndl et al. (2023a) is available at https://doi.org/10.5281/zenodo.7757034 (Maherndl et al., 2023b). HALO-(AC)[3] datasets used in this study can be accessed via the ac3airborne intake catalog (https://doi.org/10.5281/zenodo.7305585, Mech et al., 2022b). Processing routines to read IMPACTS data are available via the impacts_tools repository (https://github.com/joefinlon/impacts_tools).

## Appendix A: Microphysical overview of analyzed segments

Figure A1 (A2) presents an overview of microphysical parameters ($N_i$, $D_{32}$, $M$, IWC, LWC) observed during each analyzed IMPACTS (HALO-(AC)[3]) segment. Case study 1 (case study 2) is the fifth segment on 5 February (second segment on 1 April).

## Appendix B: Vertical distribution of $N_i$ and IWC

To investigate whether size sorting is the reason of the particle size dependency of $N_i$ and IWC variability (Sect. 4.3.1), we show vertical distributions of $N_i$ and IWC for the different size ranges in Fig. B1 and Fig. B2, respectively. Data during collocated segments is binned by their distance to CTH (as derived by radar measurements) in 100 m bins. Only bins with minimum 100 data points are shown. This leaves no data for 1.5 km below cloud top during IMPACTS. While HALO-(AC)[3]

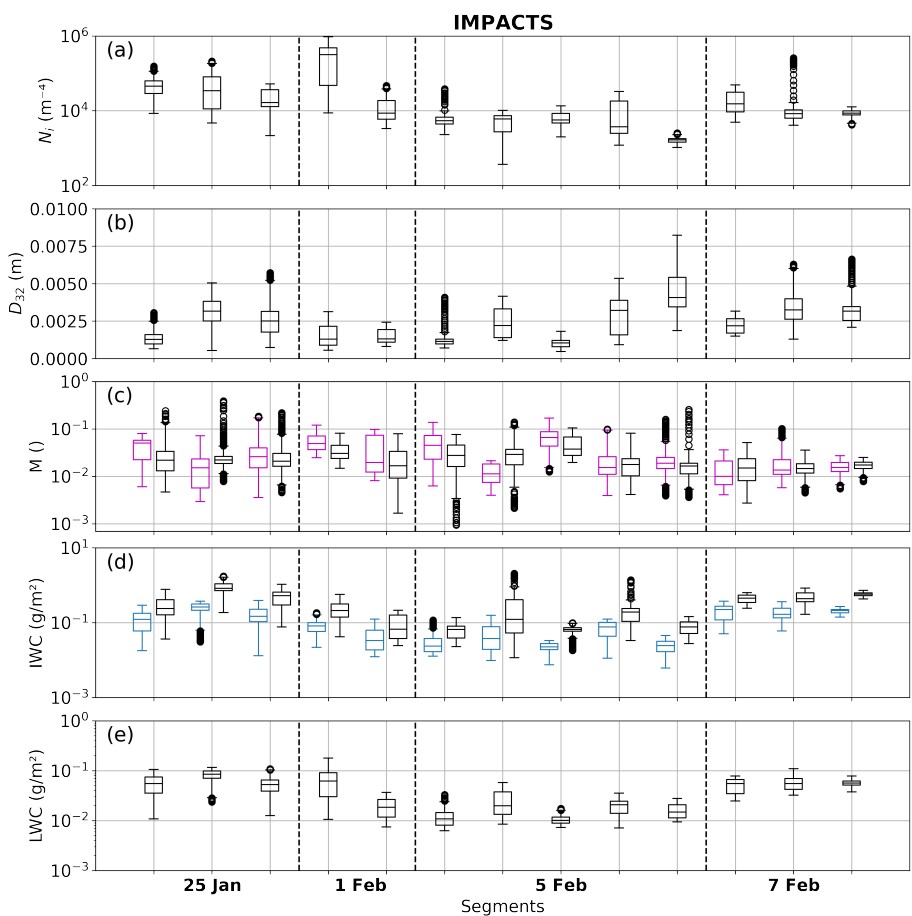

**Figure A1.** Boxplots of (a) ice number concentration $N_i$, (b) mass-weighted diameter $D_{32}$, (c) normalized rime mass $M$, (d) ice water content (IWC), and (e) liquid water content (LWC) derived during each IMPACTS segment. In (c) both combined (Ku-band) and in situ method results are shown in black and magenta, respectively. In (d) IWC is calculated accounting for riming (using combined method $M$; black) and neglecting riming ($M = 0$, blue).





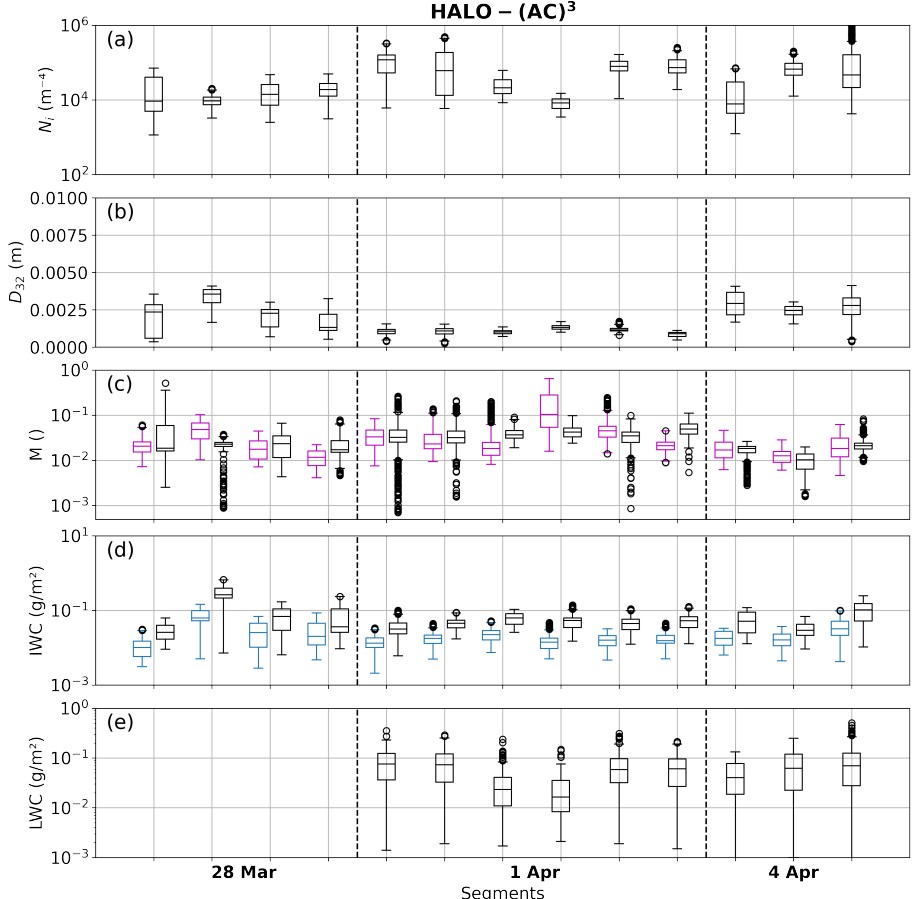

**Figure A2.** As in Fig. A1 but for HALO-(AC)$^3$ segments

data shows size sorting close to cloud top for both $N_i$ and IWC, this is not the case for IMPACTS. However, size sorting could have happened in the vertical region where we lack data. Nonetheless, $N_i$ and IWC for small particles show much larger variability during IMPACTS than during HALO-(AC)$^3$ regardless of distance to cloud top.

**Appendix C: LWP riming calculations**

This section shows the need for updrafts to explain the retrieved amounts of riming given the observed LWPs. We use simple calculations based on Fitch and Garrett (2022). Assuming a particle collects rime by falling through a liquid layer, the mass of rime accumulated can be approximated by

$$m_{rime} = A_p \, E_c \, \mathrm{LWP}, \tag{C1}$$




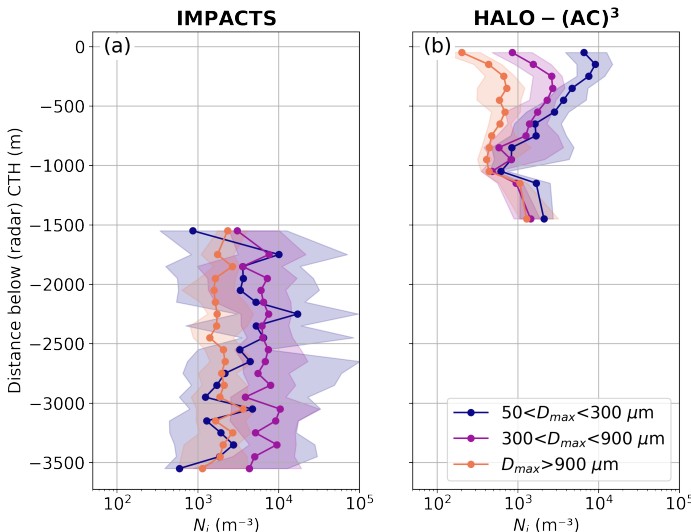

**Figure B1.** Distribution of ice number concentration $N_i$ as a function of distance to cloud top height (CTH, derived by radar) for (a) IMPACTS and (b) HALO-(AC)[3]. Lines and markers show median values; 25-75 % quantiles are shaded. Contributions of small (50-300 µm), medium (300-900 µm), and large (>900 µm) particles are shown in blue, purple, and orange.

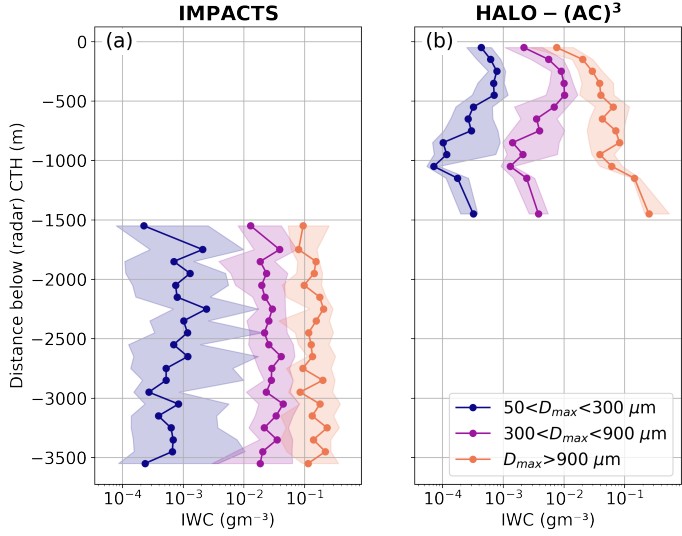

**Figure B2.** As in Fig. B1 but for ice water concent (IWC; calculated accounting for riming).



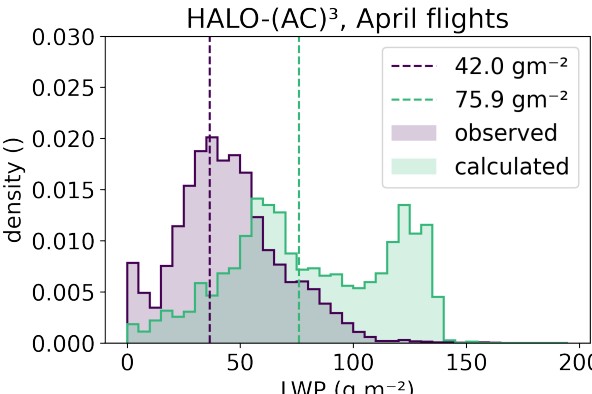

**Figure C1.** Normalized histograms of observed and calculated liquid water path (LWP) including medians (dashed lines). Observed LWP are from all 1 and 4 April data points. Calculated LWP were only derived for time steps where LWC= 0, such that it can be assumed that no further riming will take place.

where $A_p$ is the cross-sectional area of the particle, $E_c$ the combined collection and collision efficiency, and LWP the liquid water path of the liquid layer. By inserting the definition of $M$, approximating $A_p$ by a power law function of $D_{max}$ with prefactor $a_A$ and exponent $b_A$ following Maherndl et al. (2023a), and solving for LWP, we derive

$$\text{LWP} = \frac{M\, m_g}{A_p} = \frac{\pi\, \rho_g\, M}{6\, a_A(M)}\, D_{max}^{3-b_A(M)}. \tag{C2}$$

Here, $E_c$ is assumed to be 1 as a worst case estimate, although in the Arctic lower values are more realistic (Fitch and Garrett,
2022). Eq. C2 only holds for ice particles that have finished the riming process. It is therefore only applied to HALO-(AC)$^3$ data, where LWC= 0 was measured, thereby we exclude 28 March data, where LWC measurements are not available. Because ice particles occur in PSDs, we apply Eq. C2 to $D_{32}$ as a proxy for the characteristic size and the respective $M$ we retrieved for each time step. Compared to LWP observations during 1 and 4 April, the calculated LWP is much higher (Fig. C1). Therefore, it is evident, that the particles must have been exposed to the liquid layer multiple times, e.g., by cycling through up- and
downdraft regions.

*Author contributions.* NM conceptualized the study, analyzed and plotted the data, and wrote the paper. MMa contributed to the concept, acquired funding, and supervised the research project. MMo and CV collected and processed CDP, CIP, and PIP data during HALO-(AC)$^3$ and provided combined size distributions. JL collected and processed Nevzorov probe data during HALO-(AC)$^3$. IS collected and processed AMALi data during HALO-(AC)$^3$ and retrieved the CTH product. AB collected and processed CDP, Fast-CDP, 2D-S, and HVPS-3 data
during IMPACTS and provided combined size distributions. All authors reviewed and edited the draft.



*Competing interests.* The authors declare no competing interests.

*Acknowledgements.* We gratefully acknowledge funding from the Deutsche Forschungsgemeinschaft (DFG, German Research Foundation) within the framework of the Transregional Collaborative Research Center "Arctic Amplification: Climate Relevant Atmospheric and Surface Processes, and Feedback Mechanisms" project ((AC)3; grant no. 268020496–TRR 172).

Sea ice concentration data from 20 March to 10 April 2022 were obtained from https://www.meereisportal.de (last access: 2 June 2023) (grant no. REKLIM-2013-04).

We thank Mario Mech and Nils Risse from the University of Cologne for providing processed MiRAC-A and AMALi data (together with IS) as well as the retrieved LWP product during HALO-(AC)[3]. Further, we thank Christof Lüpkes and Jörg Hartmann from the Alfred Wegener Institute (AWI) for providing Polar 6 noseboom air temperature measurements. We also thank Gerald M. Heymsfield, Matthew

Walker McLinden, and Li Lihua from the NASA Goddard Space Flight Center for providing EXRAD, HIWRAP, and CRS data during IMPACTS. We are grateful to Joseph A. Finlon from the University of Washington for providing processing routines for the IMPACTS data. Further, we would like to acknowledge discussions with Matthew D. Shupe (University of Colorado and National Oceanographic and Atmospheric Administration), Heike Kalesse-Los (Leipzig University), and Patric Seifert (Leibniz Institute for Tropospheric Research) among others, whose feedback helped to shape the analysis.



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
