# Peer review of "How does riming influence the observed spatial variability of ice water in mixed-phase clouds?"

_EGUsphere, 2024_

## Referee Comment (RC1)

The authors explore the spatial variability of riming and its contribution to the clustering of ice within clouds associated with wintertime precipitation in the midlatitudes (IMPACTS) and marine cold air outbreaks over the high latitudes (HALO-(AC)[3]). They use a synergistic radar and in situ product to produce estimates of ice water content (IWC) with and without the influence of riming. By applying pairwise correlation functions to bulk microphysical parameters for a number of long flight segments (26 segments), the authors aim to capture length scales associated with IWC clustering and can further compare the functions separately between those applied to IWC including riming and those excluding riming. I particularly like the section where long swaths of in situ observations are broken up into smaller segments to essentially maximize the sample size of environments, thereby producing a more robust statistical analysis. This analysis reveals clear modes in the spatial clustering of IWC.

The paper follows logically towards its conclusions, and the figures are easily discernable and readable, and for that I thank the authors. However, I do have concerns with the robustness of the analysis. Additionally, I had confusion understanding the derived rimed and unrimed IWC, which hopefully could be better articulated/reorganized to improve upon the paper (mentioned in major comments).

I recommend this paper be reconsidered with major revisions.

Major comments:

Concerning the robustness of the results, I have two major points. First, I worry "artificial" positive pairwise correlation values are being produced by applying the moving average. This moving average is on scales of ~2 km, which is on the order of the largest observed positive correlations values (less than this value). I would propose sensitivity tests whereby varying the window size of the moving average. While I understand more robust measurements are obtained by averaging the in situ observations, it is very common to examine ice microphysical properties at 1 Hz scales (~ 100m). It would be especially prudent to use smaller windows for moving averages especially when looking at lags below a few km.

Second, there is no testing for the statistical significance of the pairwise correlation functions. This is especially a concern as standard deviations of the functions mostly overlap values equal to 0 (values expected of a homogeneously distributed system; Figure 7a,b). Further, some of the results of the rimed and "assumed-unrimed" IWC spatial inhomogeneity are nearly identical. If it's possible, applying some sort of bounds for rejection testing using white noise at some XX percentile could be helpful.

I also experienced confusion in the methodology in deriving the rimed and unrimed IWC. Concerning the organizational comment, for example, separating section 3.3 and 4.2 confused me. The derivation of IWC influenced by riming and IWC not influenced by riming seems to be separated into multiple sections (3.3 & 4.3), when section 3.1 is titled "Quantifying riming". I'm also still not sure how IWC can be separately obtained assuming riming and no riming. Are you simply using different coefficients in the mass-diameter relationship for the two variables (which I assume would be an issue since riming would in theory impact the diameter of "unrimed ice")? I'm sure it's explained in the text, however, it's difficult to determine.

Additional comments:

Line 6: delete comma after "understood"

Line 10: delete "closely" or rephrase

Line 39-40: Citation for this statement?

Line 43-44: What are the actual length scales of these smaller bands (also a citation speculating these processes would be nice).

Line 53-54: Why the long dashes?

Line 55-56: Citation showing the P3 scheme still struggles with ice processes (I get there are still broad concerns but a citation would be good when specifying a specific microphysics scheme)?

Line 63: "space-borne radar" is more commonly accepted nomenclature.

Line 65-66: should specify why measurements of IWC remain challenging (since the ensuing text implies a synergistic remote sensing/in situ method reduces uncertainty in IWC, which is misleading).

Line 94: You never define normalized rime mass. Please do.

Line 95: I'm not sure what you mean "by closure" (this was also said in the abstract). Please specify.

Line 117: rephrase "…and sampled at different frequency rates producing different spatial resolutions" or something similar.

Line 118: change "fly" to "flew"

Line 121: What does "good collocation" mean?

Line 142-143: Why is a CDP and a Fast-CDP used? Was one in error for different flights? Also, what are these probes used for exactly? Is it for PSD measurements the radar uses for calibration? Results from these probes aren't shown anywhere in the paper.

Line 149: Although understood to be somewhat common to assume >50um is all ice, it is possible droplets can get much larger than this. While the potential of icing is often the rationale for this assumption, kinetic heating of the aircraft can avoid icing at temperatures a few degrees less than 0C. In fact, I wonder if the large ice particle concentrations in IMPACTS might actually be large drops in the -5 to 0C range.

To test this by doing a temperature dependent sensitivity test, I'm curious whether results overall might be sensitive to temperature ranges (possibly not, since you do the height analysis in Appendix B, but might be worth checking).

Line 179: the collocation of radar and in situ measurements can be as far as 5 km off? That seems pretty significant based on the spatial scales you're using the pcf analysis.

Line 191-192: Perhaps I'm confused of what M really is, but isn't it possible to obtain a sum of total M over the particle population? Unsure why an average M is being obtained.

Line 202: What are synthetic rimed aggregates?

Equation 2: Are results being binned at some specified length(s) (i.e., should r be r+dr)?

Line 225: Define gaps. I'm actually unsure of what your in-cloud threshold is.

Line 257: "…and second, on spatial…"

Line 277: "…particles larger than 50 um…"

Line 280: Isn't this the definition of effective diameter (area weighted mean diameter)? Are these properties equivalent?

Line 314: "…values associated with large particles…"

Line 321: "parameters"

Line 326: Did you mean by three orders of magnitude?

Line 336-337: What are the observed ranges of M? Reflectivity seems to vary by up to 30 dBZ.

Line 337: "dBZ"

Line 343-347: What is meant by riming being "minimal" while also increasing IWC by about 2/3s?

Line 350-353: "Isn't this only true where positive values of pcf(IWCr) overlap this pcf difference?

Line 354: "…larger than zero."

Line 364: "lags"

Line 403-404: Can portions of sub-segments be resampled?

Line 411: King probe LWC results aren't shown correct? If so state it.

Line 411: "suggests" not "indicates"

Line 412: "…supersaturation with respect to ice…" although this is somewhat dependent on whether you are within that -5 to 0C range or at colder temperatures.

Line 414-416: It's difficult to tell how significant the differences are between IWCr and IWCu. I get doing the difference to highlight this but the subpanels for the respective quantities' pcfs are nearly identical (also seen in Figure 7). I think doing some statistical robustness testing would sell this point.

Line 419-420: Again, LWC is not shown correct? I get the pcf(LWC)=0 contour is shown but the modes aren't. Worth at least specifying the modes are similar.

Line 427-428: Citation?

Line 435-436: Please refer to the M panels in Figure 10.

Line 447: "deposition" not condensation.

Line 444-446 & Line 449-450: where did you show these IWC clustering results?

Line 465: "…from cloud top."

Line 480: "…larger than 6 km…"

Appendices: would be nice to keep the figures within the respective appendices.

Figure B1&C1: would be nice to show altitude or some sort of normalized cloud height rather than distance below the higher aircraft (unless this higher aircraft is essentially flying at constant altitudes).

---

## Author Comment (AC1)

**How does riming influence the observed spatial variability of ice water in mixed-phase clouds?**

N. Maherndl, M. Moser, I. Schirmacher, A. Bansemer, J. Lucke, C. Voigt, and M. Maahn

August 15, 2024

*Original Referee comments are in italic*

> manuscript text is indented, with added text underlined and

We would like to thank the reviewers for their helpful comments. We revised the manuscript and responded to all of the reviewers' comments.

In addition, we updated the IMPACTS W-band reflectivity $Z_e$ data and the normalized rime mass $M$ results obtained from the combined method for W-band, because a new version of the W-band dataset was published. $Z_e$ was adjusted downward by about $0.9\,\text{dB}$ leading to slightly lower $M$ results. The positive bias due to saturation effects for $Z_e$ values associated with large particles at W-band remains, but is less pronounced (Fig. 3c&d, Fig. 5).

**Reviewer I**

*The authors explore the spatial variability of riming and its contribution to the clustering of ice within clouds associated with wintertime precipitation in the midlatitudes (IMPACTS) and marine cold air outbreaks over the high latitudes (HALO-(AC)3). They use a synergistic radar and in situ product to produce estimates of ice water content (IWC) with and without the influence of riming. By applying pairwise correlation functions to bulk microphysical parameters for a number of long flight segments (26 segments),*

*the authors aim to capture length scales associated with IWC clustering and can further compare the functions separately between those applied to IWC including riming and those excluding riming. I particularly like the section where long swaths of in situ observations are broken up into smaller segments to essentially maximize the sample size of environments, thereby producing a more robust statistical analysis. This analysis reveals clear modes in the spatial clustering of IWC. The paper follows logically towards its conclusions, and the figures are easily discernable and readable, and for that I thank the authors. However, I do have concerns with the robustness of the analysis. Additionally, I had confusion understanding the derived rimed and unrimed IWC, which hopefully could be better articulated/reorganized to improve upon the paper (mentioned in major comments). I recommend this paper be reconsidered with major revisions.*

We thank the reviewer for the positive review and the constructive comments, which helped to improve the manuscript.

**Major comments**

*Concerning the robustness of the results, I have two major points. First, I worry "artificial" positive pairwise correlation values are being produced by applying the moving average. This moving average is on scales of $\tilde{2}$ km, which is on the order of the largest observed positive correlations values (less than this value). I would propose sensitivity tests whereby varying the window size of the moving average. While I understand more robust measurements are obtained by averaging the in situ observations, it is very common to examine ice microphysical properties at 1 Hz scales ( 100m). It would be especially prudent to use smaller windows for moving averages especially when looking at lags below a few km.*

Thank you for the comment. This is a valid concern, which we investigated using a sensitivity study as suggested. Because we calculate IWC using normalized rime mass $M$ dependent mass-size relations for each time step and we need to use running averages to get a reliable $M$ product, we can't investigate smaller running average window sizes for IWC. However, for the total number of ice particles $N_i$, we can go down to 1 Hz scales. We find that increasing the window size for computing running averages smooths peaks in the original signal and therefore the pair correlation function $\eta$ gets closer to 0 the larger the averaging window. Spatial scales where $\eta > 0$ do not change significantly as long as the window size is reasonably small. Figure R.1 and Fig. R.2 show the same analysis as Fig. 9 of the manuscript for different window sizes for IMPACTS and HALO-(AC)³, respectively. The dashed lines at 2 km and 4 km highlight the similar spacial scales where $\eta > 0$ for all window sizes, except larger 20 s for IMPACTS. We assume that due to the slow flight speed and generally lower $N_i$ during HALO-(AC)³, the 1 Hz results are noisy.

[Figure]

Figure R.1: Average pair correlation function (PCF) $\eta$ as a function of distance and lag calculated using all IMPACTS flight segments for $N_i$ for different running average window sizes. The dashed line shows the mentioned 2 km scale.

[Figure]

Figure R.2: Average pair correlation function (PCF) $\eta$ as a function of distance and lag calculated using all HALO-(AC)$^\mathbf{3}$ flight segments for $N_i$ for different running average window sizes. Here, the dashed line is drawn at 4 km.

In the revised manuscript, we added:

> Because IWC is derived using running averages of 10 s and 30 s for IMPACTS and HALO-(AC)$^\mathbf{3}$ data, respectively, we investigated the impact of the window size of the moving average on $\eta(r)$. We found that while increasing the window size from 1 s to 10 (30) s for IMPACTS (HALO-(AC)$^\mathbf{3}$) decreases absolute values of $\eta(r)$, at which lags $r$ $\eta(r)$ is positive does not change (not shown). This is because applying a moving average smooths peaks in the 1 Hz signal, but does not necessarily change their periodicity as long as the window size is reasonably small.

*Second, there is no testing for the statistical significance of the pairwise correlation functions. This is especially a concern as standard deviations of the functions mostly overlap values equal to 0 (values expected of a homogeneously distributed system; Figure 7a,b). Further, some of the results of the rimed and "assumed-unrimed" IWC spatial inhomogeneity are nearly identical. If it's possible, applying some sort of bounds for rejection testing using white noise at some XX percentile could be helpful.*

Thank you for raising this point. Not including significance testing was clearly an oversight from us. We are now using a Student's t test with a 95 % significance threshold

and reworked Fig.9. In panels (d) and (h), we only plot differences, where $\eta_{IWC}$ is positive and highlight significant positive differences with hatching. We also only show the respective $\eta = 0$ line in (a)-(c) and (e)-(g) to make the plot easier to read.

[Figure]

Figure R.3: Average pair correlation function (PCF) $\eta$ as a function of distance and lag calculated using all (a-c) IMPACTS and (e-g) HALO-(AC)$^{\mathbf{3}}$ flight segments for (a)&(e) $N_i$, (b)&(f) ice water content (IWC) accounting for riming IWC$_r$, and (c)&(g) IWC assuming no riming IWC$_u$. The Difference between (b) and (c) are shown in (d); difference between (f) and (g) in (h). Differences in (d) and (h) are only shown, where $\eta_{IWC_r} > 0$. Areas, where differences are significant according to a Student's t-test (95 % significance threshold) are hatched. $\eta = 0$ is drawn as shaded lines for the ice number concentration $N_i$ ( dash-dotted black), IWC$_r$ (solid black), IWC$_u$ (dashed black), and liquid water content (LWC,  solid blue), where LWC measurements from King probe (Nevzorov probe) measurements obtained during IMPACTS (HALO-(AC)$^{\mathbf{3}}$) are used.

In the text, we added:

Differences between positive values of IWC$_r$ and IWC$_u$ (Fig. R.3d) reveal that riming enhances the probability of ice clusters for distances larger 6 km for lags from about 1 km to 10 km (at distances of 12 km). To show the statistical significance of this enhancement, a one-sided Student's t-test with a significance threshold of 95 % is used. Areas where differences are significant are hatched (Fig. 9d).

*I also experienced confusion in the methodology in deriving the rimed and unrimed IWC. Concerning the organizational comment, for example, separating section 3.3 and 4.2 confused me. The derivation of IWC influenced by riming and IWC not influenced by riming seems to be separated into multiple sections (3.3 & 4.3), when section 3.1 is titled "Retrieving ice particle riming". I'm also still not sure how IWC can be separately obtained assuming riming and no riming. Are you simply using different coefficients in the mass-diameter relationship for the two variables (which I assume would be an issue since riming would in theory impact the diameter of "unrimed ice")? I'm sure it's explained in the text, however, it's difficult to determine.*

We apologize for the confusing structure. We have restructured the methodology section, which is now split into 3.1 "Retrieving ice particle riming", 3.2 "Deriving ice water content (IWC)", and 3.3 "Characterizing scales of ice water variability in clouds". The sensitivity study is now contained in section 4.2. Sect. 3.2 explains how we obtain IWC with and without accounting for riming in more detail, which was previously missing. We are indeed using different mass-size coefficients. When accounting for riming, we vary the mass-size coefficients depending on $M$ for each time step. When neglecting riming, we keep the coefficients fixed at values for unrimed particles. This assumes, that the particles would have the same size, if they were unrimed. As you note, this assumption is likely not realistic, because riming typically increases particle sizes, as you mention. However, with this assumption, we underestimate the increase of IWC due to riming and therefore our findings. Sect. 3.2 reads

**Deriving ice water content (IWC)**

IWC is calculated by summing the product of ice particle mass $m(D_{\max})$ and $N(D_{\max})$ for the probes' lower to upper size ranges $D_{lower}$ to $D_{upper}$

$$IWC = \sum_{D_{lower}}^{D_{upper}} m(D_{\max})N(D_{\max})\Delta D_{\max},\qquad(1)$$

where $\Delta D_{\max}$ is the size bin width. $m(D_{\max})$ is approximated by a power law relation with prefactor $a_m$ and exponent $b_m$

$$m(D_{\max}) = a_m D_{\max}^{b_m}.\qquad(2)$$

$a_m$ scales the density of ice particles (independent of particle size) and $b_m$ modulates the size dependency of particle mass, which is related to particle shape and growth processes. $a_m$ and $b_m$ depend strongly on riming (e.g., Mitchell, 1996) and reported literature values range from 0.0058 to 466

for $a_m$ and 1.8 to 3.0 for $b_m$ in SI units (e.g., discussed by Mason et al., 2018). As shown by Maherndl et al. (2023b), $a_m$ and $b_m$ strongly depend on the amount of riming, which increases particle densities. Maherndl et al. (2023b) provide $a_m$ and $b_m$ values for discrete $M$, which are interpolate to obtain parameters for a continuous $M$ in this study. We derive $a_m$ and $b_m$ for each time step as a function of the retrieved $M$. IWC is then calculated with Eq. 1 for each time step based on the measured PSD and the derived $a_m$ and $b_m$ parameters. We refer to this quantity as IWC$_r$ (IWC accounting for riming).

To estimate the contribution of the riming process to IWC, we also calculate IWC using fixed mass-size parameters $a_m$ and $b_m$ for unrimed particles (also taken from Maherndl et al., 2023b), thereby neglecting density changes (e.g., due to riming). We refer to this quantity as IWC$_u$. IWC$_u$ can be seen as the "theoretical" IWC, if the ice particles were unrimed so that the riming contribution can be estimated from the difference between IWC and IWC$_u$. However, this implies that riming does not impact the size of the unrimed ice particle, which is not necessarily the case in nature. Riming typically not only leads to an increase in ice particle density, but also ice particle size (Seifert et al., 2019). Therefore, we likely underestimate the contribution of riming to particle mass when comparing IWC$_u$ with IWC. Since we are interested in the contribution of riming to IWC variability, this approach likely results in a conservative estimate of the contribution of riming to IWC variability.

Please see the revised manuscript for additional changes (e.g., the fusion of Sect. 3.3 and Sect. 4.2 into Sect. 4.2).

**Additional comments**

*Line 6: delete comma after "understood"*

Done.

*Line 10: delete "closely" or rephrase*

Done.

*Line 39-40: Citation for this statement?*

*Line 43-44: What are the actual length scales of these smaller bands (also a citation speculating these processes would be nice).*

We rewrote the Introduction due to the comments from Reviewer 2. Therefore this paragraph is no longer included.

*Line 53-54: Why the long dashes?*

Removed.

*Line 55-56: Citation showing the P3 scheme still struggles with ice processes (I get there are still broad concerns but a citation would be good when specifying a specific microphysics scheme)?*

We included one example study:

> Their representations are therefore likely incomplete, even in sophisticated cloud microphysics schemes (e.g., Cao et al., 2023), such as the predicted particle properties (P3) scheme proposed by Morrison, Milbrandt (2015)

*Line 63: "space-borne radar" is more commonly accepted nomenclature.*

The sentence was removed in the revised introduction.

*Line 65-66: should specify why measurements of IWC remain challenging (since the ensuing text implies a synergistic remote sensing/in situ method reduces uncertainty in IWC, which is misleading).*

We rewrote the Introduction due to the comments from Reviewer 2. This should be clearer now with the following paragraphs:

> Accurate in situ measurements of IWC remain challenging (Heymsfield et al., 2010; Baumgardner et al., 2017; Tridon et al., 2019), even though in situ cloud probes can provide reliable particle size distribution (PSD) data (Korolev et al., 2013; Moser et al., 2023). Lacking IWC measurements, Deng et al. (2024) calculated IWC from PSD observations assuming that ice particle mass as a function of ice particle size follows a power law relation . Because deriving size-resolved ice particle densities from in situ PSD alone is not possible yet (to our knowledge), Deng et al. (2024) used constant mass-size parameter from Heymsfield et al. (2010). Therefore, their analyses captures IWC variability due to ice number concentration and size, but not ice particle density, which is commonly linked to riming (Erfani, Mitchell, 2017; Seifert et al., 2019)
>
> Combining collocated cloud radar and in situ PSD data allows to estimate IWC by not only showing great potential to gain better insight on microphysical processes (Nguyen et al., 2022; Mróz et al., 2021), but also to infer ice particle density changes due to riming (Maherndl et al., 2024). This way, IWC variability driven by riming-induced changes in ice particle density can

be studied. In recent years, the synergistic employment of both remote sensing and in situ instrumentation during airborne campaigns has become more common (Houze et al., 2017; McMurdie et al., 2022; Nguyen et al., 2022; Kirschler et al., 2023; Sorooshian et al., 2023; Wendisch et al., 2024; Maherndl et al., 2024).

*Line 94: You never define normalized rime mass. Please do.*

We now introduce and define the normalized rime mass $M$ in Sect. 3.1.:

> We  use the normalized rime mass $M$  (Seifert et al., 2019) to describe riming. $M$  is defined as the particle's rime mass $m_{\text{rime}}$ divided by the mass of a size-equivalent spherical graupel particle $m_g$, where we assume a rime density of $\rho_{rime} = 700$ kg m$^{-3}$:
>
> $$M = \frac{m_{\text{rime}}}{m_g} \tag{3}$$
>
> where
>
> $$m_g = \frac{\pi}{6}\rho_{\text{rime}}D_{\text{max}}^3. \tag{4}$$
>
> The maximum dimension $D_{\text{max}}$ is defined as the diameter of the smallest circle encompassing the cloud particle in m and is used to parameterize particle sizes.

*Line 95: I'm not sure what you mean "by closure" (this was also said in the abstract). Please specify.*

We removed this sentence from the introduction. Instead, we describe the method we use to retrieve riming in more detail in Sect. 3.1:

> We retrieve $M$  using the two methods introduced in Maherndl et al. (2024), which are termed the *combined method* and the *in situ method*. The methods in Maherndl et al. (2024) were developed for HALO-(AC)³, but we apply  them to IMPACTS data with slight adjustments due to different instrumentation.  In the following, we give a brief explanation of both methods and describe the adjustments for IMPACTS data. For more detail, we refer the reader to Maherndl et al. (2024).

> The combined method derives $M$ along the flight track of the in situ airplane from collocated PSD and radar reflectivity $Z_e$ measurements. It therefore relies on collocated in situ and remote sensing flights. An Optimal Estimation (Rodgers, 2000) algorithm is used to retrieve $M$ by matching simulated radar reflectivities $Z_e$ obtained from observed in situ PSD with the spatially and temporally closest measured $Z_e$. As forward operator we use the Passive and Active Microwave radiative TRAnsfer tool (PAMTRA, Mech et al., 2020) which includes empirical relationships Maherndl et al. (2023b) for estimating particle scattering properties as a function of $M$. For IMPACTS, the combined method is applied (separately) to X-, Ku-, Ka- and W-band $Z_e$ (see Sect. 4.1.3). As in Maherndl et al. (2024), we use the riming dependent mass-size parameter relation for dendrites from Maherndl et al. (2023b) that were estimated for different degrees of riming, i.e., $M$ values. Dendrites were chosen, because 86.2 % of data during the analyzed IMPACTS segments are within temperature ranges of -20 °C to -10 °C and -5 °C to 0 °C, where plate-like growth of ice crystals is preferred (only 13.8 % of the data lie between -10 °C and -5 °C, where column-like growth dominates). We assume dendrite shapes for the whole dataset, because of two reasons. First, Maherndl et al. (2024) found assuming plates or dendrites gives the same results within uncertainty estimates, and second, we want to keep the analysis of IMPACTS and HALO-(AC)³ data as consistent as possible.

In the abstract we include:

> We derive riming and IWC by combining cloud radar and in situ measurements.

*Line 117: rephrase ". . . and sampled at different frequency rates producing different spatial resolutions" or something similar.*

Done.

*Line 118: change "fly" to "flew"*

Done.

*Line 121: What does "good collocation" mean?*

We included:

> We selected these days because of the good collocation (which we define as maximum spatial offsets of 5 km and temporal offsets of 5 min; see Sect. 2.4) between the respective remote sensing and in situ aircraft as well as the data availability.

*Line 142-143: Why is a CDP and a Fast-CDP used? Was one in error for different flights? Also, what are these probes used for exactly? Is it for PSD measurements the radar uses for calibration? Results from these probes aren't shown anywhere in the paper.*

For retrieving $M$, we use combined particle size distribution (PSD) data from the respective campaign (Bansemer et al., 2022; Moser et al., 2023), which are derived from the listed instruments including a CDP and a Fast-CDP for the size range below 50 µm for HALO-(AC)[3] and IMPACTS, respectively. We assume the cloud particles in this size range to be liquid for the scattering simulations in the $M$ retrieval. We only use the Fast-CDP for IMPACTS, we removed the erroneous double-mention:

> For IMPACTS, we use data from a Fast-Cloud Droplet Probe (Fast-CDP, 2-50 µm, Lawson et al., 2017), a Two-Dimensional Stereo (2D-S, Lawson et al., 2006) probe (10-2000 µm, pixel resolution of 10 µm), one horizontally, and one vertically oriented High Volume Precipitation Spectrometer, version 3, (HVPS-3, Lawson et al., 1998)  probe (0.3-19.2 mm, pixel resolution of 150 µm).

*Line 149: Although understood to be somewhat common to assume 50um is all ice, it is possible droplets can get much larger than this. While the potential of icing is often the rationale for this assumption, kinetic heating of the aircraft can avoid icing at temperatures a few degrees less than 0C. In fact, I wonder if the large ice particle concentrations in IMPACTS might actually be large drops in the -5 to 0C range. To test this by doing a temperature dependent sensitivity test, I'm curious whether results overall might be sensitive to temperature ranges (possibly not, since you do the height analysis in Appendix B, but might be worth checking).*

We agree that analyzing the temperature or height dependence of our results would be very interesting. However, we are limited to few flight segments and performing the pair correlation analysis for given temperature bins, reduces the data amount such that results are no longer trustworthy (too little data for statistical significance). To best remove periods with large droplets from our analysis, we did the following: we only include temperatures lower -1°C. In addition, we manually looked through cloud probe images for each segment and removed two IMPACTS segments with collocated data resulting in the 13 presented segments. We previously did not state this in the manuscript and therefore included:

> As in Maherndl et al. (2024), we only include data up to -1 °C to avoid melting effects. In addition, we manually looked through in situ images of all analyzed flight segments and removed two IMPACTS segments, where we could identify supercooled droplets larger 50 µm.

*Line 179: the collocation of radar and in situ measurements can be as far as 5 km off? That seems pretty significant based on the spatial scales you're using the pcf analysis.*

Yes, 5 km is the maximum spatial offset. On average the collocation is much closer with mean offsets below 2 km for both IMPACTS and HALO-(AC)³ segments. Fig. R.4 shows histograms of the horizontal distance between in situ and radar aircraft for IMPACTS and HALO-(AC)³. We use the same collocation criteria as in Maherndl et al. (2024), where we found that the standard deviation of $Z_e$ over the average offset distances is smaller than the $Z_e$ uncertainty of 1.5 dB assumed in the $M$ retrieval (combined method). Ideally, in future studies when more collocated airborne radar and in situ data is available, the collocation criteria should be made stricter.

[Figure]

Figure R.4: Histograms of the horizontal distance in m between in situ and radar aircraft for IMPACTS (top) and HALO-(AC)³ (bottom).

*Line 191-192: Perhaps I'm confused of what M really is, but isn't it possible to obtain a sum of total M over the particle population? Unsure why an average M is being obtained.*

Using the combined method, only an reflectivity-weighted average $M$ can be obtained. We included a more detailed explanation in Sect. 3.1 (see answer to comment about line 95) and kindly refer to Maherndl et al. (2024) for more details. We hope that the inclusion of the definition of $M$ also helps.

*Line 202: What are synthetic rimed aggregates?*

Here, we meant the data set of simulated rimed aggregates from Maherndl et al. (2023a). To avoid confusion, we removed the term "synthetic" and only use "simulated" in the

revised version.

>  _Simulated rimed aggregates from Maherndl et al. (2023a)_ are used to derive empirical functions relating $\chi$ and  $D_{\max}$ to $M$, where $\chi$ and  $D_{\max}$ are derived using the same processing steps as for the respective cloud probes.

_Equation 2: Are results being binned at some specified length(s) (i.e., should r be r+dr)?_

Yes, but only when averaging $\eta$. There, we use the average flight speed of 200 m/s and 60 m/s for IMPACTS and HALO-(AC)[3] respectively to bin into 200 m and 60 m bins. We kept the equation as is, but added:

> _To perform the averaging, we bin $\eta$ into 200 m and 60 m bins for IMPACTS and HALO-(AC)[3], respectively, which corresponds to the respective distances covered in 1 s for the respective typical flight speeds._

_Line 225: Define gaps. I'm actually unsure of what your in-cloud threshold is._

We use the radar sensitivity limits to define "in-cloud", meaning that when the radar sees a signal, we assume there is a cloud. By gaps, we mean measurement gaps, i.e., NaN values for radar reflectivity or in situ PSD. We added in Sect. 4.3.2:

> In this study, only straight flight segments with a minimum of 200 s of continuous _in-cloud_ measurements are used to calculate $\eta(r)$.  _The respective radar sensitivity limits are used to define "in-cloud". We allow measurement_ gaps with a maximum length of 5 s, which are linearly interpolated.

_Line 257: "...and second, on spatial..."_

Done.

_Line 277: "...particles larger than 50 um..."_

Done.

_Line 280: Isn't this the definition of effective diameter (area weighted mean diameter)? Are these properties equivalent?_

To our knowledge, there are different definitions of effective diameter as discussed in McFarquhar & Heymsfield, (1998). Here, we use the definition of "mass-weighted" mean diameter from Maahn et al. (2015).

_Line 314: "...values associated with large particles..."_

Done.

*Line 321: "parameters"*

Done.

*Line 326: Did you mean by three orders of magnitude?*

Yes, apologies. We changed to:

By changing $D_m$ from 1 to 8 mm, IWC changes by  three orders of
magnitude.

*Line 336-337: What are the observed ranges of M? Reflectivity seems to vary by up to
30 dBZ.*

Observed ranges of M are shown in Fig. 5 and shaded in Fig.6. We added:

[Figure]

Figure R.5: Ice water content (IWC) (top), Ku-band $Z_e$ (middle), and Ka-band $Z_e$ (bottom) calculated from gamma particle size distributions as functions of $D_m$ parameter. Results for varying $N_0^*$ parameter are shown as solid and dashed lines in (a), (c), (e); for varying normalized rime mass $M$ are color-coded in (b), (d), (f). Shaded areas in (b), (d), (f), (h) indicate $M$ ranges observed during IMPACTS (90 % range: $0.005 < M < 0.15$).

*Line 337: "dBZ"*

Because we refer to differences, we mean dB not dBZ.

*Line 343-347: What is meant by riming being "minimal" while also increasing IWC by about 2/3s?*

This was misleading. We rephrased to:

> We therefore conclude that  for the

range of $M$ observed during HALO-(AC)³ and IMPACTS, the effect of riming on IWC should not be neglected  to avoid biases up to one order of magnitude for IWC.

*Line 350-353: "Isn't this only true where positive values of pcf(IWCr) overlap this pcf difference?*

Yes, we therefore now only show differences where $\eta_{IWC_r} > 0$ as discussed above.

*Line 354: ". . . larger than zero."*

Done.

*Line 364: "lags"*

Done.

*Line 403-404: Can portions of sub-segments be resampled?*

In principle, yes. However, the sampling is random.

*Line 411: King probe LWC results aren't shown correct? If so state it.*

Yes, sorry for the confusing presentation. We included:

King probe-measured LWC  cluster scales behave similarly to $N_i$ (not shown) and maximum cluster scales increase from 0.6 km to 3.0 km.

*Line 411: "suggests" not "indicates"*

Done.

*Line 412: ". . . supersaturation with respect to ice. . . " although this is somewhat dependent on whether you are within that -5 to 0C range or at colder temperatures.*

Here, we refer to the HALO-(AC)³ results, where temperatures are lower -5 °C.

*Line 414-416: It's difficult to tell how significant the differences are between IWCr and IWCu. I get doing the difference to highlight this but the subpanels for the respective quantities' pcfs are nearly identical (also seen in Figure 7). I think doing some statistical robustness testing would sell this point.*

See answer to the second major comment.

*Line 419-420: Again, LWC is not shown correct? I get the pcf(LWC)=0 contour is shown but the modes aren't. Worth at least specifying the modes are similar.*

See answer above.

*Line 427-428: Citation?*

We added:

> The lidar detects small liquid droplets at cloud top, which follow vertical motions, therefore leading to higher CTH in updraft regions (Abel et al., 2017)
> .

*Line 435-436: Please refer to the M panels in Figure 10.*

We included:

> Given that the least (most) amount of riming (Fig. 10c,f,i) occurred on 4 (1) April, we conclude that in the studied MCAO clouds mesoscale updraft features likely enhance riming at spatial scales of 3-5 km.

*Line 447: "deposition" not condensation.*

Thanks, we changed to:

> Ice particles grow through  depositional growth and riming, which leads to enhanced probabilities of ice clusters at these scales.

*Line 444-446 & Line 449-450: where did you show these IWC clustering results?*

These results are shown in Fig. 9h and discussed in Sect. 4.3.2. Fig. 9h shows that riming influences IWC clustering at two spatial scales: 1. riming increases the probability of clustering at scales below 1-2 km, in the same range as the roll cloud circulation and updraft features found by Schirmacher, et al. (2024), and 2. riming leads to additional clustering at 3-5 km. In the revised manuscript, we added a reference to the figure:

> In the presence of additional mesoscale updraft features, IWC clusters also occur at spatial scales of 3-5 km (Fig. 9h).

*Line 465: ". . . from cloud top."*

Done.

*Line 480: ". . . larger than 6 km..."*

Done.

*Appendices: would be nice to keep the figures within the respective appendices. Figure B1&C1: would be nice to show altitude or some sort of normalized cloud height rather*

*than distance below the higher aircraft (unless this higher aircraft is essentially flying at constant altitudes).*

The figures are now within their respective appendix sections. Regarding Fig. B1 and C1: Here, we do show the distance to cloud top, not to the higher aircraft. We include the "radar" in brackets, because here we use cloud top height as derived by radar.

**Additional References**

McFarquhar, G. M., and A. J. Heymsfield, 1998: The Definition and Significance of an Effective Radius for Ice Clouds. J. Atmos. Sci., 55, 2039–2052, https://doi.org/10.1175/1520-0469(1998)055<2039:TDASOA>2.0.CO;2.

---

## Author Comment (AC2)

**How does riming influence the observed spatial variability of ice water in mixed-phase clouds?**

N. Maherndl, M. Moser, I. Schirmacher, A. Bansemer, J. Lucke, C. Voigt, and M. Maahn

August 15, 2024

*Original Referee comments are in italic*

> manuscript text is indented, with added text underlined and

We would like to thank the reviewers for their helpful comments. We revised the manuscript and responded to all of the reviewers' comments.

In addition, we updated the IMPACTS W-band reflectivity $Z_e$ data and the normalized rime mass $M$ results obtained from the combined method for W-band, because a new version of the W-band dataset was published. $Z_e$ was adjusted downward by about $0.9\,\mathrm{dB}$ leading to slightly lower $M$ results. The positive bias due to saturation effects for $Z_e$ values associated with large particles at W-band remains, but is less pronounced (Fig. 3c&d, Fig. 5).

**Reviewer II**

*The authors using aircraft observations investigate the spatial variability of riming and its contribution to the clustering of ice within clouds. They employ observations from IMPACTS and HALOAC3 where collocated aircraft observations are unique for the analysis. Although I feel this work has some new contributions to the community, the current presentation and organization need major revisions. More detailed comments may be given after addressing these key issues. Sorrying for the delayed post, I tried to under-*

*stand this work by reading it for multiple times, but it is very frustrating in interpreting its core logics and idea.*

We thank the reviewer for the comments, which helped to improve the manuscript.

**Major comments**

*1. The research motivation is poorly structured. The introduction just lists all relevant topics, from spatial distribution of MPC, properties of MPC in mid-latitudes and Arctic, to the data you have, and then riming. I do not see a clear logic and strong motivation for this research. Frankly, it left me the impression that you did the analysis just because you have these data available.*

*Also, the introduction omits very key details on the unique datasets from IMPACTS and HALO-(AC)3. So far, we have so many aircraft observations, why just these two campaigns fit your study?*

We have rewritten the introduction highlighting why we need the unique collocated radar and in situ data obtained during HALO-(AC)$^3$ and IMPACTS to study riming and its impact on IWC variability. The paragraphs on MPC in mid-latitudes and the Arctic were redundant and therefore removed. Please see the revised manuscript and the file with marked changes for the new version of the introduction.

*2. Causality issue. Although the analysis and results are no doubt interesting, I question the statements in many places such as riming enhances the probability of IWC clusters. What I can expect is some dynamical mechanisms such as the generating cells as discussed influence the IWC clustering, and such mechanisms influence riming and clustering. You may say that IWC clusters at certain scales are rimed, but I do not agree with the reasoning that the clustering of ice (macrophysics) is influenced by riming (microphysics).*

*Similar causal issues apply to many statements on the relationship between riming and IWC. In many places, the authors state that riming affects/influence IWC variability, however, these is no given evidence showing the IWC variability is due to riming. Riming is one of the characteristics of the ice clusters, not the factor leading to the variability.*

We want to address this comment together with the next comment, because we fear misunderstandings arose due to poor structuring and lack of detailed information on methodology on our side. We apologize for that.

*3. Some key methods lack details or are arbitrary.*

*(1) Quantifying riming. After reading several times of the section 3.1, I have no idea*

*what is the combined method for quantifying riming. I am frustrated in understanding the logic.*

*(2) Quantifying IWC variability. This key statement at L350 lacks physics background. The logic of interpreting signs of is straightforward. However, it is arbitrary to interpret the sign of 1 – 2 in the same way, since a positive 1 – 2 can be the results of two negative .*

In response to (1): The "combined method" for quantifying riming was introduced in Maherndl et al. (2024), we therefore only cited this paper without going into detail. In hindsight, we agree that a more detailed explanation is needed to follow our methodology and logic. We therefore have rewritten the Methods section (Sect. 3), which is now split into Sect. 3.1 "Retrieving ice particle riming", 3.2 "Deriving ice water content (IWC)", and 3.3 "Characterizing scales of ice water variability in clouds". To better explain the combined method we rewrote Sect. 3.1 to:

**Retrieving ice particle riming**

[revised manuscript text omitted]

Because there is a separate paper describing the method in detail, we refer to Maherndl et al. (2024), if the Reviewer is interested in more information.

We also added Sect. 3.2 "Deriving ice water content (IWC)" to explain how we derive IWC with and without accounting for riming:

**Deriving ice water content (IWC)**

IWC is calculated by summing the product of ice particle mass $m(D_{\max})$ and $N(D_{\max})$ for the probes' lower to upper size ranges $D_{lower}$ to $D_{upper}$

$$IWC = \sum_{D_{lower}}^{D_{upper}} m(D_{\max})N(D_{\max})\Delta D_{\max}, \tag{4}$$

where $\Delta D_{\max}$ is the size bin width. $m(D_{\max})$ is approximated by a power law relation with prefactor $a_m$ and exponent $b_m$

$$m(D_{\max}) = a_m D_{\max}^{b_m}. \tag{5}$$

$a_m$ scales the density of ice particles (independent of particle size) and $b_m$ modulates the size dependency of particle mass, which is related to particle shape and growth processes. $a_m$ and $b_m$ depend strongly on riming (e.g., Mitchell, 1996) and reported literature values range from 0.0058 to 466 for $a_m$ and 1.8 to 3.0 for $b_m$ in SI units (e.g., discussed by Mason et al., 2018) . As shown by Maherndl et al. (2023b), $a_m$ and $b_m$ strongly depend on the amount of riming, which increases particle densities. Maherndl et al. (2023b) provide $a_m$ and $b_m$ values for discrete $M$, which are interpolate to obtain parameters for a continuous $M$ in this study. We derive $a_m$ and $b_m$ for each time step as a function of the retrieved $M$. IWC is then calculated with Eq. 4 for each time step based on the measured PSD and the derived $a_m$ and $b_m$ parameters. We refer to this quantity as $IWC_r$ (IWC accounting for riming).

> To estimate the contribution of the riming process to IWC, we also calculate IWC using fixed mass-size parameters $a_m$ and $b_m$ for unrimed particles (also taken from Maherndl et al., 2023b), thereby neglecting density changes (e.g., due to riming). We refer to this quantity as $\text{IWC}_u$. $\text{IWC}_u$ can be seen as the "theoretical" IWC, if the ice particles were unrimed so that the riming contribution can be estimated from the difference between IWC and $\text{IWC}_u$. However, this implies that riming does not impact the size of the unrimed ice particle, which is not necessarily the case in nature. Riming typically not only leads to an increase in ice particle density, but also ice particle size (Seifert et al., 2019). Therefore, we likely underestimate the contribution of riming to particle mass when comparing $\text{IWC}_u$ with IWC. Since we are interested in the contribution of riming to IWC variability, this approach likely results in a conservative estimate of the contribution of riming to IWC variability.

We rephrased the section headline "Characterizing scales of cloud variability" to "Characterizing scales of ice water variability in clouds". We are analyzing the variability of IWC in clouds (microphysics) not the variability of cloud cover (macrophysics). We apologize in case this was not made sufficiently clear. We included:

> Similar to Deng et al. (2024), we use the pair correlation function (PCF) to quantify the spatial inhomogeneity of ice water in the observed clouds.

Now we would like to finally address the comment about causality, which is of course a valid issue to raise. Changes in IWC can be linked to three reasons, 1. an increase/decrease in ice particle number, 2. and increase/decrease in ice particle size, and 3. an increase/decrease in ice particle density, which can all occur at once. Here, we assume changes in ice particle density are linked to riming, because rimed particles typically have higher densities. We don't discuss what causes changes in ice particle number or size, except for a few speculations regarding differences in IMPACTS and HALO-(AC)³ data. Because we have in situ PSD observations for both campaigns, we can analyze changes in IWC due to changes in number and size, while neglecting changes in IWC due to changes in ice particle density by assuming constant mass-size parameter in the calculation of IWC (as was done in Deng et al. (2024)). Thanks to the unique collocated cloud radar and in situ measurements during IMPACTS and HALO-(AC)³, we are able to retrieve the normalized rime mass $M$ for each time step using the combined method. Knowing $M$ allows us to vary mass-size parameter for each time step. We can therefore calculate IWC accounting for density changes due to riming. If we then compare scales of variability between IWC neglecting ($\text{IWC}_u$)and IWC accounting for riming ($\text{IWC}_r$), we can single out the influence of riming on IWC variability. The only difference in the variability of $\text{IWC}_r$ and $\text{IWC}_u$ is variability due to riming-induced density changes. We therefore argue that stating riming influences IWC variability at scales where we see differences in $\text{IWC}_r$ and $\text{IWC}_u$ variability is reasonable. In the revised manuscript, these explanations are given in the new Sect. 3.2 (see answer above)

and the core logic is introduced in the introduction:

> Accurate in situ measurements of IWC remain challenging (Heymsfield et al., 2010; Baumgardner et al., 2017; Tridon et al., 2019), even though in situ cloud probes can provide reliable particle size distribution (PSD) data (Korolev et al., 2013; Moser et al., 2023). Lacking IWC measurements, Deng et al. (2024) calculated IWC from PSD observations assuming that ice particle mass as a function of ice particle size follows a power law relation . Because deriving size-resolved ice particle densities from in situ PSD alone is not possible yet (to our knowledge), Deng et al. (2024) used constant mass-size parameter from Heymsfield et al. (2010). Therefore, their analyses captures IWC variability due to ice number concentration and size, but not ice particle density, which is commonly linked to riming (Erfani, Mitchell, 2017; Seifert et al., 2019)
>
> Combining collocated cloud radar and in situ PSD data allows to estimate IWC by not only showing great potential to gain better insight on microphysical processes (Nguyen et al., 2022; Mróz et al., 2021), but also to infer ice particle density changes due to riming (Maherndl et al., 2024). This way, IWC variability driven by riming-induced changes in ice particle density can be studied. In recent years, the synergistic employment of both remote sensing and in situ instrumentation during airborne campaigns has become more common (Houze et al., 2017; McMurdie et al., 2022; Nguyen et al., 2022; Kirschler et al., 2023; Sorooshian et al., 2023; Wendisch et al., 2024; Maherndl et al., 2024).

Regarding (2), we agree and apologize for not making this clearer in the original manuscript. We reworked Fig. 9, where we only plot differences, where $\eta_{IWC}$ is positive. In addition, we highlight significant positive differences with hatching using a Student's t test with a 95 % significance threshold. We also only show the respective $\eta = 0$ line in (a)-(c) and (e)-(g) to make the plot easier to read.

[Figure]

Figure R.1: Average pair correlation function (PCF) $\eta$ as a function of distance and lag calculated using all (a-c) IMPACTS and (e-g) HALO-(AC)$^3$ flight segments for (a)&(e) $N_i$, (b)&(f) ice water content (IWC) accounting for riming IWC$_r$, and (c)&(g) IWC assuming no riming IWC$_u$. The Difference between (b) and (c) are shown in (d); difference between (f) and (g) in (h).  $\eta = 0$ is drawn as shaded lines for the ice number concentration $N_i$ ( dash-dotted black), IWC$_r$ (solid black), IWC$_u$ (dashed black), and liquid water content (LWC,  solid blue), where LWC measurements from King  (Nevzorov probe measurements obtained during IMPACTS (HALO-(AC)$^3$) are used.

In the text, we added:

Differences between  IWC$_r$ and IWC$_u$ (Fig. 9d) reveal that riming enhances the probability of ice clusters for distances larger 6 km for lags from about 1 km to 10 km (at distances of 12 km).

*4. Lacks of in-depth analysis of the observations from the two campaigns. The authors tried to use data from two campaigns for the analysis. However, I do no see clear physics explaining the observed differences between the two campaigns, nor general conclusions given.*

The aim of our work was to identify spatial scales at which riming leads to high IWC, i.e., ice clustering. Using pair correlation functions, we show that accounting for riming increases to probability of ice clustering as opposed to not accounting for riming in both mid-latitude (IMPACTS) and Arctic (HALO-(AC)³) cloud cases that we studied. This clustering occurs at the same spatial scales whether accounting for riming or not and at the same spatial scales as liquid water content (LWC) clustering. We therefore argue, that riming is enhanced in cloud regions with high amounts of LWC leading to high IWC. Because high IWC and high LWC occur at the same spatial scales also when neglecting riming, we hypothesize that in these regions supersaturation is high with respect to both liquid and ice.

In the Arctic cold air outbreak clouds observed during HALO-(AC)³, we found an additional interesting feature not present in the mid-latitude cases. At spatial scales of 3-5 km, ice clustering occurs in IWC accounting for riming, which is not present when neglecting riming. We therefore argue, that at these 3-5 km scales, riming causes high IWC (ice clustering). However, this feature can not be explained by enhanced LWC. We investigated the physical explanation, why more riming occurs at these scales and found evidence that mesoscale updrafts are responsible. We hypothesize that in these stronger updrafts, ice particles are suspended longer in the air, before the fall out thereby having more time to rime even if there is not more liquid water present. In the conclusions, we summarize these findings highlighting the differences between both campaigns related to ice clustering.

A more detailed analysis of both campaigns is outside the scope of this work and already covered in other publications. McMurdie et al. (2022) and Wendisch et al. (2024) give overviews of the respective campaigns; the synoptic conditions during HALO-(AC)³ are analyzed in Walbröl et al. (2024).

**Minor comments**

*L28&29 I understand that there are very few studies on the spatial distributions of ice and liquid mass, but I do not appreciate inappropriate citations. The listed ones have discussed the impacts of different cloud phases, they did not mention the spatial distribution.*

The original sentence read "Their spatial distribution **as well as ice and liquid mass**..." therefore not only referring to the spatial distribution. We wanted to refer to the fact that how much liquid vs. how much ice composes the cloud and how it is distributed spatially impacts the listed properties as shown in the cited studies. However, we agree that the sentence was lengthy and easy to misread. We rephrased to:

> Mass and the ratio of ice and liquid  particles play a critical role not only in precipitation processes, but also cloud lifetime, radiative budget

(Sun, Shine, 1994; Shupe, Intrieri, 2004; Turner, 2005), and climate feedbacks (Choi et al., 2014; Bjordal et al., 2020).

Due to restructuring the introduction, we discuss the spatial distribution of ice particles and liquid droplets in a later paragraph.

*L43 Literature should be given. Also, I do not agree the statement. Smaller-scale bands are mostly linked to dynamics and associated microphysics.*

Due to restructuring the motivation, we removed this paragraph.

*Section 3.3 Sensitivity study is poorly structured. It is difficult to capture the logic in the present format. It seems that this section was splitted into two parts. It is recommended to combine section 3.3 and section 4.2.*

We apologize for the poor structure. We merged sections 3.3 and 4.2 into one. The section now reads:

**Sensitivity study**

To motivate our further analysis and to evaluate whether the retrieved amounts of riming significantly impact IWC, we conduct a sensitivity study.

We assume that $N(D_{\max})$ follows a modified gamma distribution and use the normalized form introduced by Delanoë et al. (2005, 2014) and extended by Maahn et al. (2015) for the maximum dimension $D_{\max}$

$$N(D_{\max}) = N_0^* \frac{(b_m + \mu + 1)^{b_m+\mu+1}\Gamma(b_m + 1)}{\Gamma(b_m + \mu + 1)(b_m + 1)^{b_m+1)}} \left(\frac{D_{\max}}{D_m}\right)^{\mu} e^{-(b_m+\mu+1)D_{\max}/D_m},$$

(6)

where $N_0^*$ is the overall scaling parameter, $\mu$ the shape parameter, and $D_m$ is the "mass-weighted" scaling parameter for the particle size. We vary $N_0^*$ and $D_m$ — which can be calculated from PSD moments (see Maahn et al., 2015) — based on 10 to 90% quantile values derived from all measured PSDs during IMPACTS. Exclusively IMPACTS data was chosen, because larger particles and higher number concentrations were measured during IMPACTS than during HALO-(AC)³. $\mu$ is varied from 0 to 64 based on extreme values reported in the literature (Tridon et al., 2022). $M$ is varied from 0.005 to 1, which correspond to the 10 % quantile of $M$ retrieval results from both campaigns and the maximum "physical" $M$ based on its definition.

We find that although median $M$ are below 0.03 for both campaigns, even small amounts of riming — or rather changes in ice particle density — can

result in large changes of IWC. Figure R.2 shows IWC calculations assuming gamma PSDs with varying $N_0^*$ (left column) and $M$ (right column) as a function of $D_m$. Similar to Maahn, Löhnert (2017), we find the shape parameter $\mu$ does not impact IWC or $Z_e$ significantly and therefore only $\mu = 0$ is shown. $D_m$, which can be seen as a proxy for particle size, has the largest impact on IWC. By changing $D_m$ from 1 to 8 mm, IWC changes by  three orders of magnitude. IWC increases by about one order of magnitude, when $N_0^*$ — the proxy for total number concentration of particles — is increased by one order of magnitude. Depending on $D_m$, varying $M$ can result in IWC changes up to two order of magnitudes. When only considering $M$ values encountered during the analyzed campaigns, the change in IWC reaches one order of magnitude.

To show the impact of riming on radar reflectivity $Z_e$ — which can be seen as a proxy for IWC —, we conduct a sensitivity study for Ku and Ka-band $Z_e$. In doing so, we aim to highlight the importance of accounting for riming in radar retrievals. $Z_e$ is forward simulated using the same PSDs with  PAMTRA assuming a temperature of $-10$ °C. Particle scattering is parameterized with the riming-dependent parameterization (Maherndl et al., 2023b). X-band is not shown due to being nearly identical to Ku-band; W-band is not shown due to the riming-dependent parameterization bias for large $D_m$ at W-band (see Sect. 4.1.3). Varying $M$ within observed ranges results in $Z_e$ changes of up to 20 dB depending on $D_m$ for both Ku- and Ka-band, albeit with slightly larger spread at Ka-band. Similar to Fig. R.2, varying $D_m$ results in the largest $Z_e$ changes. Observed ranges of $M$ result in larger $Z_e$ changes than observed ranges of $N_0^*$. Therefore in our data set, $Z_e$ depends more heavily on riming than on number concentration.

We therefore conclude that  for the range of $M$ observed during HALO-(AC)³ and IMPACTS, the effect of riming on IWC should not be neglected and can cause biases up to one order of magnitude.

[Figure]

Figure R.2: Ice water content (IWC) (top), Ku-band $Z_e$ (middle), and Ka-band $Z_e$ (bottom) calculated from gamma particle size distributions as functions of $D_m$ parameter. Results for varying $N_0^*$ parameter are shown as solid and dashed lines in (a), (c), (e); for varying normalized rime mass $M$ are color-coded in (b), (d), (f). Shaded areas in (b), (d), (f), (h) indicate $M$ ranges observed during IMPACTS (90 % range: 0.005 < M < 0.15).

*L263 Awkward logic. MPC properties vary between IMPACTS and HALO-(AC)3 just because of different synoptic situations (Sect. 2.3) and measurement locations? This is very misleading.*

Sorry, this was misleading. We rephrased to:

>  MPC properties, synoptic situations (Sect. 2.3), and measurement locations (Fig. 1)  vary between IMPACTS and HALO-(AC)³.

*Figure 5. Why the boxplot of W-band retrieval in IMPACTS is different from others? You did not explain it in the caption.*

We are unsure, what you are referring to. The caption of Fig. 5 reads "W-band results during IMPACTS are dashed due to biases (see text)." The explanation is given in the text (see lines 325-333 in the revised manuscript), because a discussion of results is typically not included in figure captions.

[Figure]

Figure R.3: Box plots and superimposed violin plots showing normalized rime mass $M$ results obtained from a closure of collocated radar reflectivity $Z_e$ and in situ particle size distribution ("combined method" from Maherndl et al. (2024)) for radar reflectivities available during (a) IMPACTS and (b) HALO-(AC)[3]. W-band results during IMPACTS are dashed due to biases (see text). $M < 0.01$ are plotted at 0.01 to be visible on the logarithmic scale.

*L394 this conclusion lacks evidence.*

We use $\eta$ to study IWC variability as e.g., in Deng et al. (2024). $\eta_{IWC}$ is a proxy for the probability of IWC clustering. The only difference between the variability of $IWC_u$ and $IWC_r$ is variability due to riming-induced density changes. We therefore argue that it is reasonable to say that differences between $\eta$ computed for $IWC_u$ and $IWC_r$ are linked to riming.

*Figure 11. Observations from two campaigns were analyzed, but you only show the conceptual diagram for HALOAC3. This is something of an anticlimax.*

We show the conceptual diagram only for HALO-(AC)[3], because we found that riming leads to IWC clustering at additional spatial scales as opposed to the mid-latitude clouds observed during IMPACTS. We found that riming does not change spatial scales of ice clustering for the IMPACTS cases. Both of these findings have not been reported in the literature before (to our knowledge). Figure 11 should act as a visual aid to understand the different spatial scales of ice clustering we identified for HALO-(AC)[3] as well as the impact of riming on these scales. Because we did not find that riming changes spatial scales of ice clustering during IMPACTS, we don't think that an additional figure is necessary. To better highlight the findings for IMPACTS, we included:

In the analyzed segments of winter storm clouds measured during IMPACTS, IWC clusters occur at spatial scales smaller than about 3 km for segment distances of 15 km. Accounting for riming enhances ice cluster probabilities (Fig. R.1d). However, riming does not lead to significantly enhanced occurrences of IWC clusters at other scales. LWC clusters for segment distances of 15 km occur at the same spatial scales of about 3 km as clusters of $N_i$. Therefore, liquid droplets and ice particles are likely formed together in regions with supersaturation with respect to liquid and ice. Because LWC clusters and the IWC cluster enhancement through riming occur at similar spatial scales, we hypothesize that LWC variability (at least in part) drives riming. By increasing IWC, riming leads to enhanced probabilities of IWC clusters for IMPACTS.

---

## Referee Report (RR1)

The revised manuscript is far improved in its readability and overall quality, and the authors addressed most of my points. However, I still have minor comments which need addressing. Line numbers are in reference to the tracked changes document.

Major comment: I have still found a significant number of grammatical and punctuation issues. I sincerely urge the authors to take time diligently proofreading the manuscript and editing appropriately.

Figure 11: you should move it up before the conclusion section.

Line 72-75: Are you saying there is no method to obtain the mass of an ice PSD beyond utilizing Dmax and Nice? There have been efforts made to quantify ice particle mass accounting for their habit (e.g., McFarquhar et al. 2019; University of Illinois OAP software). In fact, you mention that am and bm can be modulated as a function of particle habit (line 284-285). Perhaps I'm misunderstanding.

Line 123: You deleted where you introduced MCAO. In fact, this occurs for multiple acronyms in the paper (including NASA and DFG). Please thoroughly proofread and write out terms for every acronym.

Line 121-124: Correct the numbering of these bullet points.

Line 141-143: Be consistent in "Section" vs "Sect."

195-197: Why did you set a threshold of -1C degree to account for kinetic heating of the aircraft allowing for the ability to avoid de-icing and thus sampling these large drops? This heating can occur at a few degrees below zero (perhaps include a source). Otherwise, why even include this threshold? Since you said you looked through the in situ particle imagery.

Line 203: Here and I believe other places you say "typical" with quotation marks. Remove them.

Line 276-277: I'm unsure why you're introducing this in situ method only to show it in the case studies? Why include this at all?

I also ask this, since in the case studies you reference it but you never say why you don't use it versus the dual platform method. Why not use this method and avoid the uncertainties associated with collocating the remote sensing observations?

Line 313: Perhaps I missed it, but what are the radar sensitivity limits? Include them.

Line 403: This should not be titled Campaign overview. It is not that. Call it something like "Additional riming product discussion"

Line 411: This seems out of nowhere. I don't know what x is.

Line 418-419: I don't get how this further motivates your study. It seems this is showing the range of expected M and associated Ze. If so, rewrite this introductory sentence accordingly.

Line 515-518: In the comments to the reviewers, you confirm resampling with replacement. State this in the manuscript.

Line 569: "convective cells"

Line 570-571: "liquid condensation" change to "condensational growth"

Line 600-601: Rewrite sentence.

Line 621: "models' representations of MPCs".

---

## Author Response (AR2)

**How does riming influence the observed spatial variability of ice water in mixed-phase clouds?**

N. Maherndl, M. Moser, I. Schirmacher, A. Bansemer, J. Lucke, C. Voigt, and M. Maahn

October 10, 2024

*Original Referee comments are in italic*

> manuscript text is indented, with added text underlined and

We revised the manuscript and responded to all of the reviewer's comments.

**Reviewer I**

*The revised manuscript is far improved in its readability and overall quality, and the authors addressed most of my points. However, I still have minor comments which need addressing. Line numbers are in reference to the tracked changes document.*

We thank the reviewer for taking the time to review our manuscript for a second time. Thank you for the constructive comments, which helped to improve the manuscript.

**Major comment**

*I have still found a significant number of grammatical and punctuation issues. I sincerely urge the authors to take time diligently proofreading the manuscript and editing appropriately.*

Thank you for addressing this issue. We carefully proofread the manuscript and corrected all grammatical and punctuation issues we could find. We also deployed the help of writing assistance tools.

**Additional comments**

*Figure 11: you should move it up before the conclusion section.*

There is an issue with the LaTeX diff tool that we used to compile the author's tracked changes document. In the revised version of the manuscript (without tracked changes), the figure appears before the conclusion.

*Line 72-75: Are you saying there is no method to obtain the mass of an ice PSD beyond utilizing Dmax and Nice? There have been efforts made to quantify ice particle mass accounting for their habit (e.g., McFarquhar et al. 2019; University of Illinois OAP software). In fact, you mention that am and bm can be modulated as a function of particle habit (line 284-285). Perhaps I'm misunderstanding.*

No, we intended to make the argument that with no further information expect the PSD (so no information about particle shape, IWC, etc.), it is not possible to derive particle density. In retrospect, this argument does not make much sense, because most (if not all) cloud probes that derive PSDs also provide particle images. We rephrased to:

> relationship. Because it is difficult to derive size-resolved ice particle densities from in situ observations alone, Deng et al. (2024) used constant mass-size  parameters from Heymsfield et al. (2010).

*Line 123: You deleted where you introduced MCAO. In fact, this occurs for multiple acronyms in the paper (including NASA and DFG). Please thoroughly proofread and write out terms for every acronym.*

Thank you. We added:

> The main objective of the HALO-(AC)³ campaign was  to study Arctic air mass transformations during warm air intrusions and marine cold air outbreaks (MCAOs).

Also for DFG and NASA:

> The  German Research Foundation (DFG) funded field campaign HALO-(AC)³ ...

> The Investigation of Microphysics and Precipitation for Atlantic Coast-Threatening

Snowstorms (IMPACTS, McMurdie et al., 2022) campaign was a  National Aeronautics and Space Administration (NASA) sponsored field campaign ...

We also wrote out CMIP6, HALO-(AC)³, LWP, and MODIS:

> ... Coupled Model Intercomparison Project version 6 (CMIP6) ...

> ... HALO-(AC)³ (Wendisch et al., 2024; HALO, High Altitude and Long Range Research Aircraft – (AC)³ Project on Arctic Amplification Climate Relevant Atmospheric and Surface Processes and Feedback Mechanisms; see https://halo-ac3.de/, last access: 8 October 2024)...

> During HALO-(AC)³, brightness temperature $T_B$ measurements at 89 GHz were collected and are used to derive the liquid water path (LWP).

> According to the level-2 Moderate-resolution Imaging Spectroradiometer (MODIS) ...

*Line 121-124: Correct the numbering of these bullet points.*

This is again an issue with LaTeX diff that does not occur in the revised manuscript (without tracked changes).

*Line 141-143: Be consistent in "Section" vs "Sect."*

Done.

*195-197: Why did you set a threshold of -1C degree to account for kinetic heating of the aircraft allowing for the ability to avoid de-icing and thus sampling these large drops? This heating can occur at a few degrees below zero (perhaps include a source). Otherwise, why even include this threshold? Since you said you looked through the in situ particle imagery.*

Because we wanted to avoid partially melted ice particles, which would not be represented well in our scattering simulations, we had to set a temperature threshold to sort out all flight segments with too warm temperatures. We thought this threshold would be sufficient. However, further inspection of in situ images during the remaining flight segments showed that some segments with large drops were still left. We removed these segments from the analysis. Setting no temperature threshold would mean unnecessary additional work, i.e. looking through images for segments with temperatures larger 0°C.

We added:

> As in Maherndl et al. (2024), we only include data up to -1 °C to avoid melting ice particles, which are not represented well in the scattering

simulations that we perform. In addition, we manually looked through in situ images of all  remaining flight segments ...

*Line 203: Here and I believe other places you say "typical" with quotation marks. Remove them.*

Done.

*Line 276-277: I'm unsure why you're introducing this in situ method only to show it in the case studies? Why include this at all? I also ask this, since in the case studies you reference it but you never say why you don't use it versus the dual platform method. Why not use this method and avoid the uncertainties associated with collocating the remote sensing observations?*

We introduced the in situ method for reference and rough uncertainty estimation. We believe that showing results for both methods in the case studies and in the overview in Appendix A gives more credit to the combined method $M$ that we use in the further analysis. It also helps to better show the uncertainty of our riming product. We trust the combined method results more than the in situ method because the former uses the full PSD. For the in situ method we are left with a size gap where $M$ cannot be derived due to instrument resolution. This can be problematic if there are heavily rimed particles that occur exactly in this size range (see Maherndl et al., 2024, for an example of such a case).

Because this was not explained well in the text, we added:

> Only a subset of ice particles can be used to derive $M$ with the in situ method, because particles cannot touch edges to derive $P$ and  must be large enough to derive meaningful $\chi$.  Because of these two criteria, ice particles with $D_{max}$ in the range of about 1.0-1.4 mm and 2.0-6.0 mm are neglected by the in situ method when using the HALO-(AC)³ and IMPACTS particle probes, respectively. Therefore, we assume that the combined method—which uses the full PSD—gives more reliable results  if the aircraft are reasonably collocated, as shown in Maherndl et al. (2024) for HALO-(AC)³. We use $M$ derived with the combined method for all further analysis steps. For reference and uncertainty estimation, we show the in situ method $M$  in Sect. 4.1.1 and 4.1.2 and in Appendix A.

*Line 313: Perhaps I missed it, but what are the radar sensitivity limits? Include them.*

We included:

> EXRAD, HIWRAP Ku-band, HIWRAP Ka-band, and CRS have sensitivity

limits of -15 dBZ, 0 dBZ, -5 dBZ, and -28 dBZ at 10 km range, respectively.
... MiRAC-A has a sensitivity limit of about -40 dBZ at 3 km range.

*Line 403: This should not be titled Campaign overview. It is not that. Call it something like "Additional riming product discussion"*

We changed the title of this section to "Riming product statistics and discussion".

*Line 411: This seems out of nowhere. I don't know what x is.*

We rephrased the paragraph so that the size parameter $x$ is properly introduced:

> For IMPACTS, the  discrepancy between the W-band results  and the other frequency bands is due to the  occurrence of large ice particle sizes. Because of saturation effects, the riming-dependent parameterization (Maherndl et al., 2023)  has a positive $Z_e$ bias for  large relative sizes of scattering particles. The relative size of a scattering particle is defined by its size parameter $x = 2\pi\alpha_e D_{max}/\lambda$, where $\alpha_e$ is the  effective aspect ratio of the ice particle, and $\lambda$ the radar wavelength. Positive biases occur for $x > 4$. The positive $Z_e$ bias for $x > 4$ results in a positive bias of $M$.

*Line 418-419: I don't get how this further motivates your study. It seems this is showing the range of expected M and associated Ze. If so, rewrite this introductory sentence accordingly.*

We agree that the phrasing was not ideal. We changed to:

> To  show the effect of expected $M$ on $Z_e$ and to evaluate whether the retrieved amounts of riming significantly impact IWC, we conduct a sensitivity study.

*Line 515-518: In the comments to the reviewers, you confirm resampling with replacement. State this in the manuscript.*

We added:

> This is repeated 100 times and the average $\eta$ over all (sub)segments of the respective campaign is calculated. In principle, parts of sub-segments can be resampled. However, the sampling process is random. To perform the averaging, ...

*Line 569: "convective cells"*

Done.

*Line 570-571: "liquid condensation" change to "condensational growth"*

Done.

*Line 600-601: Rewrite sentence.*

We rephrased to:

> During IMPACTS,  the maximum spatial scales of $N_i$, IWC, and LWC clusters inside clouds are 0.6-3 km for distances of  2-15 km.

*Line 621: "models' representations of MPCs".*

Done.